# Graphene collage on Ni-rich layered oxide cathodes for advanced lithium-ion batteries

Chang Won Park [1,2,5], Jung-Hun Lee [3,5], Jae Kwon Seo[3], Won Young Jo[3], Dongmok Whang[1,3], Soo Min Hwang [3✉] & Young-Jun Kim [3,4✉]

The energy storage performance of lithium-ion batteries (LIBs) depends on the electrode capacity and electrode/cell design parameters, which have previously been addressed separately, leading to a failure in practical implementation. Here, we show how conformal graphene (Gr) coating on Ni-rich oxides enables the fabrication of highly packed cathodes containing a high content of active material (~99 wt%) without conventional conducting agents. With 99 wt% $LiNi_{0.8}Co_{0.15}Al_{0.05}O_2$ (NCA) and electrode density of ~4.3 g cm$^{-3}$, the Gr-coated NCA cathode delivers a high areal capacity, ~5.4 mAh cm$^{-2}$ (~38% increase) and high volumetric capacity, ~863 mAh cm$^{-3}$ (~34% increase) at a current rate of 0.2 C (~1.1 mA cm$^{-2}$); this surpasses the bare electrode approaching a commercial level of electrode setting (96 wt% NCA; ~3.3 g cm$^{-3}$). Our findings offer a combinatorial avenue for materials engineering and electrode design toward advanced LIB cathodes.

[1] School of Advanced Materials Science and Engineering, Sungkyunkwan University, Suwon, Republic of Korea. [2] AEB)Cell Development Team, Samsung SDI Co., LTD, Yongin, Republic of Korea. [3] SKKU Advanced Institute of Nano Technology (SAINT), Sungkyunkwan University, Suwon, Republic of Korea. [4] Department of Nano Engineering, Sungkyunkwan University, Suwon, Republic of Korea. [5]These authors contributed equally: Chang Won Park, Jung-Hun Lee. ✉email: smhwang@skku.edu; yjkim68@skku.edu

Substantial advancements in battery technologies have driven societal transformations and innovation. Li-ion batteries (LIBs) are the undeniable frontrunners that have powered various portable electronic devices and electric vehicles; they have also provided large-scale energy storage platforms for the efficient use of renewable energies[1–3]. The consumer markets still push toward improved energy and power density at affordable costs[4,5]. The energy storage capability of LIB cells depends on the electrode/cell design and electrode materials[5,6]. Typically, an increase in the cell energy results from an increased areal capacity ($Q_{areal}$) by increasing the electrode thickness (areal mass loading, $m_{areal}$ of active materials). However, such electrode thickening deteriorates high-rate (power) performance, due to the increased charge (electron and Li-ion) transport distance within the electrode[7,8], and volumetric energy density. The electrodes in the state-of-the-art LIB cells employ few percentages (wt%) of inactive materials, such as large-surface-area carbonaceous conducting agents and polymeric binders to ensure the conductive percolation network and mechanical endurance, respectively. Such use of nanoscale conductive additives, typically carbon black (CB), necessitates an equivalent amount (wt%) of insulating binders, and causes parasitic reactions with organic electrolyte during cell operation[9,10], which remain invariable obstacles in electrode design toward improving cell performance.

Here, we report an effective method for improving the energy density of electrodes by reducing the content of inactive components through conformal graphene (Gr) coating on active materials. High-nickel layered oxide cathodes, such as $LiNi_{0.8}Co_{0.15}Al_{0.05}O_2$ (NCA) and $LiNi_{0.8}Co_{0.1}Mn_{0.1}$ (NCM811), are examined as model electrodes, since they are currently promising cathodes for high-energy density LIBs[11,12]. We use a dispersion of electrochemically exfoliated Gr nanosheets that are functionalized with an amphiphilic surfactant acting as glue for Gr coating on the surface of Ni-rich oxides; this is distinct from previous Gr coating approaches using solution-processed reduced graphene oxide (rGO)[13–15] and mechanical milling[16,17]. By eliminating the conventional conductive additive (CB) and minimizing the binder content, we demonstrate highly dense Ni-rich cathodes (99 wt% NCA; electrode density ($\rho$) ~4.3 g cm$^{-3}$) with a high areal capacity ($Q_{areal}$) of ~5.4 mAh cm$^{-2}$ (~38% increase) and high volumetric capacity ($Q_{vol}$) of ~863 mAh cm$^{-3}$ (~34% increase) at current rate of 0.2 C (~1.1 mA cm$^{-2}$), relative to the bare electrode with a commercial level of electrode setting (96 wt% NCA, $\rho$ ~3.3 g cm$^{-3}$).

## Results

### Electrochemical exfoliation of Gr and coating on Ni-rich oxides.
Gr coatings have been employed to bestow a conducting surface on various active materials in LIBs[13–19]. Most studies have relied on solution-processed rGO[13–15], CVD-Gr[17], or mechanical milling processes[16,17], which typically require cumbersome and high-temperature processing and damage the host materials. To circumvent such disadvantages, we used Gr nanosheets fabricated through electrochemical exfoliation with commercial graphite foils[20,21]. The detailed experimental setup and exfoliation behavior are described in the "Methods" section and in Supplementary Fig. 1. The exfoliated Gr nanosheets possess lateral sizes of 0.1–17 μm and a mean thickness of 3.3 nm (Supplementary Fig. 2a–c). The Gr edges are partially oxygen-functionalized (C–OH/C–O/C=O) due to the oxidation of graphite by OH$^-$ ions during the electrochemical process (Supplementary Fig. 3)[21].

The surface chemistry of Gr nanosheets is decisive for coating on target materials. The hydrophobicity of the basal plane impedes conformal Gr wrapping via plane-to-plane

contact with Ni-rich oxide particles, the surface of which has hydrophilic lithium hydroxides and/or lithium carbonates caused by exposure to the atmosphere[22,23]. It has also been reported that for graphitic carbon, Li-ion diffusion across the basal plane is sluggish (~10$^{-11}$ cm$^2$ s$^{-1}$), relative to that along the basal plane (~10$^{-6}$ cm$^2$ s$^{-1}$)[24], implying that Gr coatings hinder Li-ion mass transfer between the electrolyte and active materials upon charging and discharging of Gr-coated Ni-rich cathodes. Typically, stacked Gr layers allow the ion transport mainly through the defects/holes in the sheet plane and the boundaries between Gr sheets[24–26]. The higher-order defects, such as divacancies and holes, with low diffusion barrier heights are preferred for facile Li-ion transport[25–27]. To resolve these issues, we employed an amphiphilic surfactant, 1,2-distearoyl-sn-glycero-3-phosphoethanolamine-N-[methoxy(polyethylene-glycol) (DSPE-mPEG), as glue for Gr coating, together with a sonication to reduce Gr size and defects generation[28,29]. By adding an appropriate amount of DSPE-mPEG into the dispersion of exfoliated Gr in N,N-dimethylformamide (DMF, 1 wt% Gr) and subsequent high-power tip sonication, we prepared a stable, functionalized Gr dispersion with reduced lateral sizes (<500 nm; Supplementary Figs. 2d–f and 4). Finally, Gr-coated Ni-rich oxide powder was obtained by simple vortex mixing of the Gr solution containing Ni-rich oxide particles and centrifugation. Figure 1a illustrates a schematic of the Gr coating procedure, in which a graphite foil was electrochemically exfoliated into Gr nanosheets and then ripped to small Gr pieces by sonication and functionalization. The Gr scraps were attached onto Ni-rich oxide particles in a face-to-face manner, owing to the adhesive amphiphilic surfactant, whose hydrophilic head binds to the oxide surface, while retaining π–π interactions between the basal plane of Gr and the hydrophobic tails[27]. The repeated coating process allows almost conformal attachment of the Gr scraps on the surface of Ni-rich oxides; this attachment resembles the "collage" technique of art creation.

Figure 1b–d displays the electron micrographs of the NCA particle that was subjected to the three times (3T)-coating process. The particle surface was found to be entirely covered with Gr nanosheets (Fig. 1b, c), which was clearly distinct from the surface of bare NCA particles (Fig. 1e and Supplementary Fig. 5). The transmission electron microscopy (TEM) observation revealed that ~3.1-nm-thick Gr nanosheets were attached along the NCA surface and possessed an interplanar spacing of ~3.5 Å (Fig. 1d). Such Gr layers are easily found on the particle surface, overall (see Supplementary Fig. 6). We also carried out confocal Raman mapping on the Gr-coated particles to check the surface coverage of Gr on NCA (see Fig. 1f–i). Although a part of the particles is out of focus due to their rough surface morphology, the Gr map (G band, Fig. 1h; red color) was well-matched with the NCA map ($E_g$ and $A_{1g}$ modes, Fig. 1g; blue color), indicating good coverage of the Gr on NCA particles. The coating process without DSPE-mPEG resulted in mediocre attachment of Gr on the surface of NCA particles, even after 3T-coating (Supplementary Fig. 7). Based on thermogravimetric analysis (TGA) data, the Gr content in the coated (3T) NCA particles is evaluated to be ~0.5 wt% (Supplementary Fig. 8). The electrical conductivity of the Gr-coated NCA powder was measured using four-point probes by varying the applied pressure, and the resulting data are plotted as a function of powder density (Fig. 1j). For comparison, commercial CB powder and bare NCA powders mixed with two different contents of CB (98:1 and 96:2 in wt%) were also tested; the Gr-coated NCA powder exhibited a lower electrical conductivity than the CB powder and mixture powders with the value exceeding that of bare NCA powder by more than three orders of magnitude. The measured data were fitted using the

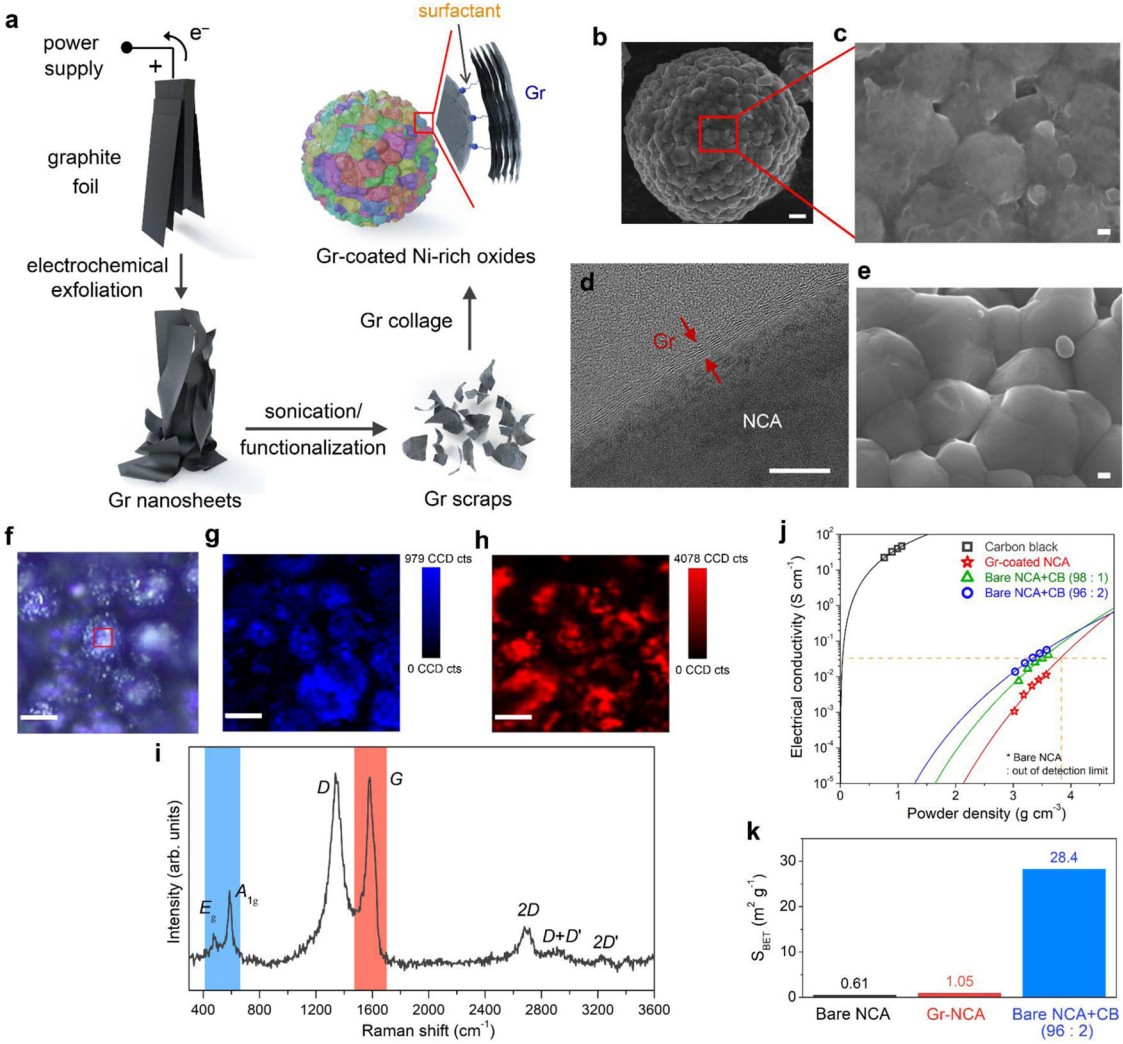

**Fig. 1 Gr coating on Ni-rich oxides. a** Schematic of Gr collage on Ni-rich oxide particles. SEM images of **b**, **c** Gr-coated and **e** bare NCA particles and **d** HRTEM image of Gr-coated NCA particle. The scale bar in **b**–**e** indicates 1 µm, 100 nm, 10 nm, and 100 nm, respectively. **f** Optical and **g**, **h** confocal Raman mapping images and **i** Raman spectrum on Gr-coated NCA particles. The Raman maps of NCA ($E_g$ and $A_{1g}$ mode) and Gr (G band) are shown in **g** and **h**, respectively. The scale bars in **f**–**h** indicate 4 µm. **j** Electrical conductivity of Gr-coated NCA powder and mixture powders containing NCA and CB with weight ratios of 96:2 and 98:1. **k** BET surface area of the bare and Gr-coated NCA powders and the mixture powder (NCA:CB = 96:2).

percolation scale law ($\sigma = \sigma_0 (\emptyset - \emptyset_c)^n$, for $\emptyset > \emptyset_c$, refer to Supplementary Note 1). Considering that commercial Ni-rich cathodes employ $\rho$ ~3.3 g cm$^{-3}$ and their electrode composition (~96 wt% active material), the Gr-coated NCA electrodes were expected to have comparable electrical conductivities above ~3.8 g cm$^{-3}$, which will be discussed below. These results indicate that our solution process enables conformal Gr coating on NCA particles, owing to the amphiphilic surfactant. Compared with the mixture powder containing the bare NCA and CB (NCA: CB = 96:2 in wt%), the Gr-coated NCA powder has a surface area ($S_{BET}$) as small as that of the bare NCA (Fig. 1k and Supplementary Fig. 9), possibly reducing side reactions stemming from large-surface-area conductive agents during cell operation[9,10].

We determined the presence of DSPE-mPEG on the Gr nanosheets and in Gr-coated NCA particles by spectroscopic analyses. The bare and Gr-coated NCA powders were compared using Fourier transfer infrared (FT-IR) analysis; the results revealed the presence of DSPE-mPEG (–CH$_3$ and phosphate groups) within the coated particles (Supplementary Fig. 10). Raman mapping data also suggested that the Gr sheets after

sonication of Gr dispersion containing DSPE-mPEG show notably high D/G ratios locally, i.e., higher defect densities, compared with the as-exfoliated Gr sheets (Supplementary Fig. 11). We further investigated the presence of the DSPE-mPEG surfactant by fluorescence microscopy (FM), which is typically used to observe biomaterials that selectively interact with fluorescence dyes. Considering that the DSPE-mPEG surfactant contains an anionic phosphate group, we used a cationic dye of rhodamine B (RhB) that fluoresces bright-red (refer to the orange boxes in Fig. 2g). Firstly, we examined the as-exfoliated Gr sheets, which were functionalized with different contents of DSPE-mPEG (0–0.2 wt%; refer to "Methods" section), to confirm the attachment of DSPE-mPEG on Gr sheets. The DSPE-mPEG/Gr solutions were diluted with a RhB solution and spread on SiO$_2$/Si wafer. The FM images of the as-exfoliated and functionalized Gr samples are shown in Supplementary Fig. 12. Graphitic carbon has been reported to quench the fluorescence from dye molecules adsorbed on its surfaces via the excited-state energy transfer, significantly decreasing the fluorescence intensity[30–32]. The as-exfoliated Gr sheets appeared dark due to the effective quenching. After the functionalization process with higher contents of DSPE-

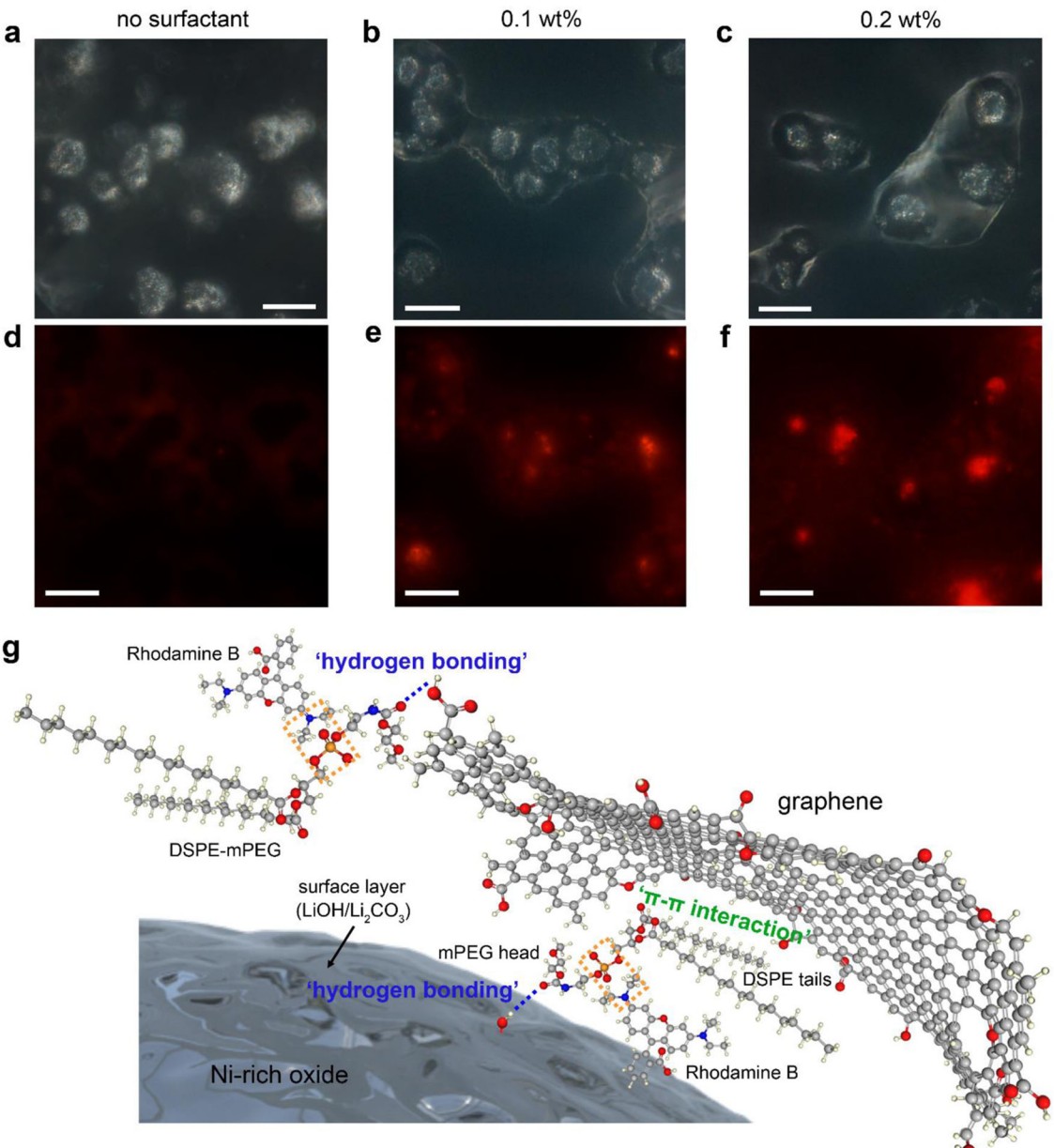

**Fig. 2 FM images of Gr-coated NCA particles and the bonding configuration between Gr and Ni-rich oxide. a–c** Optical and **d–f** FM images of NCA particles that were subjected to Gr coating process (one time) **a**, **d** without the DSPE-mPEG surfactant and using the surfactant of **b**, **e** 0.1 wt% and **c**, **f** 0.2 wt%, placed on a carbon adhesive tape. The scale bars in **a–f** indicate 20 μm. **g** Schematic illustration of the bonding configuration between Gr and Ni-rich oxide. The Gr layer, DSPE-mPEG, and rhodamine B are illustrated using a ball-and-stick model (drew by Marvin Sketch), where red, gray, ivory, blue, and orange colors indicate oxygen, carbon, hydrogen, nitrogen, and phosphorus, respectively. The DSPE-mPEG could bind to the oxide surface by the hydrophilic head via "hydrogen bonding" with the hydroxyl groups of oxide surface, while retaining "π–π interactions" between the basal plane of Gr and the hydrophobic tails. In addition, the surfactant could bind to Gr edges by the hydrophilic head via "hydrogen bonding" with the hydroxyl functional groups of Gr, consequently contributing to conformal Gr coating in a face-to-face manner. The orange dotted boxes highlight the possible reactions between the cationic rhodamine B and the anionic phosphate group of DSPE-mPEG.

mPEG, the Gr sheets fluoresce appeared red from their edge regions (see Supplementary Fig. 12c, d), indicating considerably weakened quenching ability, owing to the presence of dye-adsorbed DSPE-mPEG at the edges.

Next, FM was performed for the NCA particles that were subjected to Gr coating process (one time) with and without the use of DSPE-mPEG. As shown in Fig. 2a, d, overall the NCA particles processed without the DSPE-mPEG showed a dark contrast; their surroundings appeared weak red due to remnant RhB salts, which could have been incorporated during the washing step. On the other hand, the bright-red color was clearly observed on the NCA particles coated using DSPE-mPEG (Fig. 2b, e and  c, f). These results confirmed the presence of surfactant in the Gr-coated particles. We speculated that the surfactant interacts on the basal plane with the hydrophobic tails in the DSPE, while being attached on the NCA surface via hydrogen bonding between the surface hydroxyl group (–OH) and the hydrophilic mPEG head (see Fig. 2g and Supplementary Fig. 13)[22,27]. In addition, the surfactant could bind to the Gr edges by the hydrophilic head via hydrogen bonding with the

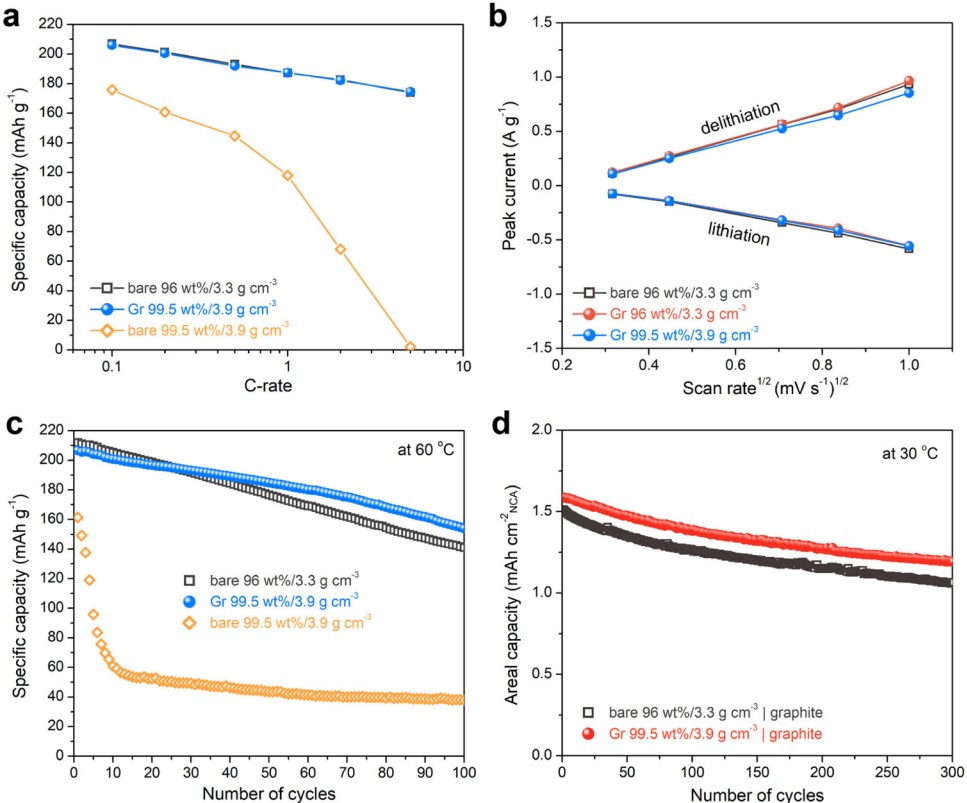

**Fig. 3 Electrochemical performance of Gr-coated NCA cathodes. a** Rate capability of the Gr-coated and bare electrodes at 0.1–5 C. **b** Relationship between the peak current ($i_p$) and the square root of the scan rate ($\nu^{1/2}$) for lithiation/delithiation. **c** Discharge capacity of the Gr-coated and bare NCA electrodes, measured at 60 °C in the voltage range of 2.75–4.3 V vs. $Li^+$/Li at 0.5 C. The bare and Gr-coated electrodes have different electrode parameters of $\varphi = 96:2:2$ vs. $\varphi = 99.5:0:0.5$ and $\rho$ ~3.3 g cm$^{-3}$ vs. $\rho$ ~3.9 g cm$^{-3}$. **d** Areal capacity of full cells made with the NCA cathode and graphite anode during 300 cycles, measured at 30 °C in the voltage range of 2.75–4.2 V at 0.5 C.

hydroxyl functional groups of the Gr, consequently contributing to a conformal Gr coating on NCA in a face-to-face manner. This conjecture is partly corroborated by controlled experiments on different types of surfactants (refer to Supplementary Note 2 and Supplementary Fig. 14).

**Electrochemical properties of Gr-coated Ni-rich cathodes.** We evaluated the electrochemical properties of Gr-coated NCA cathodes. By optimizing the Gr coating conditions, including the surfactant content, Gr multicoating, and binder content (Supplementary Note 3 and Supplementary Figs. 15–17), we fabricated the Gr-coated NCA electrode without using conventional conducting agents (NCA:CB:PVdF ($\varphi$) = 99.5:0:0.5; $\rho$ ~3.9 g cm$^{-3}$). Our Gr-coated NCA electrode showed good rate and cycling performance comparable to that of the bare electrode with a commercial level of electrode parameters ($\varphi = 96:2:2$; $\rho$ ~3.3 g cm$^{-3}$). As shown in Fig. 3a, the rate capabilities of the bare and Gr-coated NCA electrodes, which were measured in the voltage range of 2.75–4.3 V vs. $Li^+$/Li at 0.1–5 C, were similar. For comparison, the bare electrode with electrode parameters of $\varphi = 99.5:0:0.5$ and $\rho$ ~3.9 g cm$^{-3}$ is also shown. These results reveal that the coated Gr nanosheets form sufficient conductive percolation networks within the electrodes even without the use of CB. This is also corroborated by electrochemical impedance spectroscopy (EIS) results on as-assembled cells with NCA electrodes before the formation process. In the Nyquist plots (Supplementary Fig. 18), the bare and Gr-coated NCA electrodes show one semicircle, which is mainly due to the contact resistance between the NCA particles and current collector and the interparticle resistance

within the electrodes[33,34]. This indicates that the Gr coating diminishes the contact resistance, comparable to that of the bare electrode ($\varphi = 96:2:2$; $\rho$ ~3.3 g cm$^{-3}$), which is denoted as the counterpart for comparison unless otherwise indicated. As mentioned above, Gr nanosheets coated on NCA particles possibly impede the Li-ion diffusion between the electrolyte and the active material during cell operation. We performed cyclic voltammetry (CV) on the Gr-coated electrode and compared the $Li^+$ diffusion kinetics with those of bare electrodes, based on the Randles–Sevcik equation ($i_p \propto A \cdot D_{Li^+}^{0.5} \cdot \nu^{0.5}$), where the peak current ($i_p$) is proportional to the surface area of redox-active materials ($A$), $Li^+$ diffusion coefficient ($D_{Li^+}^{0.5}$), and scan rate upon CV measurements ($\nu^{0.5}$); thus, the slope ($\frac{i_p}{\nu^{0.5}}$) denotes the $Li^+$ diffusion property (Supplementary Fig. 19). The Gr-coated NCA electrode and the bare electrode exhibited almost similar kinetics for Li-ion diffusion upon charge (delithiation) and discharge (lithiation; Fig. 3b). The comparison between the bare and Gr-coated electrodes with the same electrode parameters ($\varphi = 96:2:2$; $\rho$ ~3.3 g cm$^{-3}$) confirms a negligible effect of the Gr coating on the Li-ion mass transfer.

Cycle performance of the Gr-coated NCA electrodes was examined at 30 and 60 °C. The Gr-coated electrode displays similar cycling stability during 100 cycles at 30 °C, as compared to the bare electrode (Supplementary Fig. 20); on contrast, the capacity retention at 60 °C is slightly better than that of bare one (Fig. 3c). For comparison, the bare electrode without CB (99.5 wt% NCA; $\rho$ ~3.9 g cm$^{-3}$) was also tested at 60 °C; a drastic capacity fading during the initial few cycles was observed. The Nyquist plots of the bare and Gr-coated samples

displayed similar increasing trend in the semicircle (due to the film resistance; $R_f$ and charge-transfer resistance; $R_{ct}$) before and after 100 cycles (Supplementary Fig. 21). In addition, we analyzed the TEM-electron energy loss spectroscopy (EELS) observations of the cycled NCA particles, which were prepared by focused-ion-beam cross-sectioning. As shown in Supplementary Fig. 22, both the bare and Gr-coated NCA particles have a NiO-like structure in the vicinity of the surface of the primary particles after cycling[35,36]. However, the Gr-coated particles showed lesser intraparticle cracking than the bare particles, which can be attributed to the protection by Gr from electrolyte penetration into the secondary particle interior to some extent during repeated cycles[6,11,12,37].

We compared the full-cell performance of the bare and the Gr-coated NCA cathodes. The full cells were assembled by pairing the cathodes with graphite anodes at an N/P ratio of 1.1 and cycled at 30 °C in the voltage range of 2.75–4.2 V at a current rate of 0.5 C (Fig. 3d and Supplementary Fig. 23). The two full cells exhibited similar capacity retention (>70%) during 300 cycles. The cell employing the Gr-coated cathode delivered a slightly higher $Q_{areal}$ of ~1.19 mAh cm$^{-2}$ NCA than that of the cell using the bare cathode (~1.06 mAh cm$^{-2}$ NCA) at the 300th cycle. This high value was obtained because of using a higher NCA content (99.5 wt%), while minimizing inactive components (only 0.5 wt% PVdF) within the electrode, thus indicating the efficacy of the Gr coating on NCA particles.

The NCM811 cathode, another Ni-rich layered oxide material, was examined to prove the effects of the Gr collage on the electrochemical performance. The NCM particles were coated thrice using a Gr dispersion (1 wt% Gr) containing DSPE-mPEG (0.1 wt%) in the same manner. As shown in Supplementary Fig. 24, the Gr sheets were attached uniformly throughout the surfaces of the NCM particles. The rate and cycle performance of the coated NCM electrode with $\varphi = 99.5{:}0{:}0.5$ and $\rho$ ~3.9 g cm$^{-3}$ was compared to that of the bare NCM electrode ($\varphi = 96{:}2{:}2$; $\rho$ ~3.3 g cm$^{-3}$). Despite the absence of CB, the coated electrode exhibited good rate properties at 0.2–10 C and stable cycle performance at 60 °C, which are comparable to those of the bare electrode (Supplementary Fig. 25). There was no significant difference in the Nyquist plots between the bare and coated electrodes before and after 100 cycles. In addition, we coated LiCoO$_2$ (LCO) particles with Gr. Interestingly, the Gr coverage and coating behavior was mediocre, compared to the cases of Ni-rich oxides (NCA and NCM811); several Gr sheets agglomerated and the degree of Gr attachment after the rinsing step was not adequate (Supplementary Fig. 26). The resulting rate capability of the coated LCO electrode was not as good as the bare one. The differences in the coating behaviors between the LCO and Ni-rich oxides could be attributed to their different surface characteristics; the surface of the latter shows higher basicity than that of the former, being prone to reacting with H$_2$O/CO$_2$ upon exposure to ambient air and forming LiOH/Li$_2$CO$_3$ (refs. [22,38]). Thus, the surface feature of Ni-rich oxides contributes to conformal Gr coverage through its interaction with the DSPE-mPEG via hydrogen bonding, as discussed above.

Ni-rich, layered oxide cathodes suffer from cycling and thermal instabilities arising from the phase transformation from layered to spinel/rock-salt phases. This transition is accompanied by evolution of gaseous O$_2$/CO$_2$ and dissolution of transition metal (TM) ions during the repeated cycles and accelerated during high-voltage charging (>4.3 V vs. Li$^+$/Li) and/or under elevated temperatures when in the charged (delithiated) state[11,12,37,39]. Accordingly, we evaluated the Gr coating effect on the high-voltage cycling and thermal stabilities, as well as the extent of TM-ion dissolution and gas evolution from the Gr-coated NCA electrode. Figure 4a depicts the discharge capacity retention of the

bare and Gr-coated NCA electrodes measured at 0.5 C and 30 °C. There was no significant difference in the voltage profiles and resulting capacity retention between the bare and coated electrodes during the 150 cycles (refer to Supplementary Fig. 27). Next, the bare and Gr-coated NCA electrodes were charged to 4.3 V vs. Li$^+$/Li and then washed with dimethyl carbonate (DMC), followed by immersion in a 1 M LiPF$_6$ electrolyte at 60 °C for 1 week. The TM ions dissolved in the electrolyte were quantitatively estimated using inductively coupled plasma optical emission spectrometry (ICP-OES). Fewer TM ions (Ni and Co) were detected in the Gr-coated electrode compared to the bare electrode (Fig. 4b), which could be attributed to the protective effect of the Gr layer coated on the NCA particles. Figure 4c presents the differential scanning calorimetry (DSC) profiles of the bare and Gr-coated electrodes that were subjected to full charging, displaying almost similar, sharp peaks. This was due to the exothermic decomposition via phase transformation from the initial layered (rhombohedral; space group: $R\bar{3}m$) structure into the disordered spinel (space group: $Fd\bar{3}m$) structure, involving the concurrent release of O$_2$ gas[39]. Nevertheless, the Gr-coated electrode showed a slightly delayed onset of the exothermic reaction. This phenomenon could be attributed to incorporation of a few inactive components that could release CO$_2$ through their oxidation reactions with the highly active oxygen species (such as O$_2^-$, O$^-$, and O$_2^{2-}$) released from the charged NCA particles[39]. These findings are also supported by the results of the gas-cell experiments at 60 °C. An appropriate amount of the electrolyte was injected into the chamber of a home-made gas-cell containing a fully charged electrode (refer to Supplementary Fig. 28). Assuming that the pressure increase is primarily due to gas evolution from the NCA electrode, the Gr-coated electrode showed lesser gas evolution during the test than the bare electrode (Fig. 4d). These results demonstrate that the Gr coating, and hence the incorporation of few inactive components has positive effects on the cycling and thermal stabilities of NCA electrodes.

**Electrode engineering toward achieving high volumetric energies.** A high $Q_{areal}$ of electrodes is achieved by increasing the $m_{areal}$ of active materials; however, this leads to electrode thickening, thereby decreasing the power performance[7,8,40]. Thus, thin but dense electrode architectures are preferred for enhanced volumetric energy and power density. We prepared two different types of Gr-coated, dense NCA electrodes ($\rho$ ~4.3 g cm$^{-3}$) without CB ($\varphi = 99{:}0{:}1$); one electrode (denoted as "Gr-A") had the same $m_{areal}$ of ~20 mg cm$^{-2}$ and the other (denoted as "Gr-B") had a similar thickness of ~63 μm ($m_{areal}$ ~27 mg cm$^{-2}$), for comparison with a ~60-μm-thick bare electrode, approaching a commercial level of electrode parameters ($m_{areal}$ ~20 mg cm$^{-2}$; $\rho$ ~3.3 g cm$^{-3}$; and $\varphi = 96{:}2{:}2$). Figure 5a–c shows the cross-sectional scanning electron microscopy (SEM) images of the three electrodes with different electrode parameters. Interestingly, the Gr-coated electrodes have good structural integrity without notable inter-/intraparticle cracking, even after undergoing high-pressure calendaring for such a high electrode density (Fig. 5a, b and Supplementary Fig. 29). In contrast, the bare electrode shows local microcracks (Fig. 5c and Supplementary Fig. 29), as observed in previous studies[37,41,42]. These observations confirm that Gr coatings on NCA particles mitigate the pressure exerted during calendaring, while CB nanoparticles damage the NCA particles under high pressures.

The discharge rate capabilities of the three NCA electrodes were evaluated at current rates of 0.1–5 C, and the resulting $Q_{areal}$ and $Q_{vol}$ plots are depicted in Fig. 5d, e, respectively (refer to Supplementary Fig. 30). All the electrodes exhibit monotonic capacity decays with the increasing areal current. The Gr-B

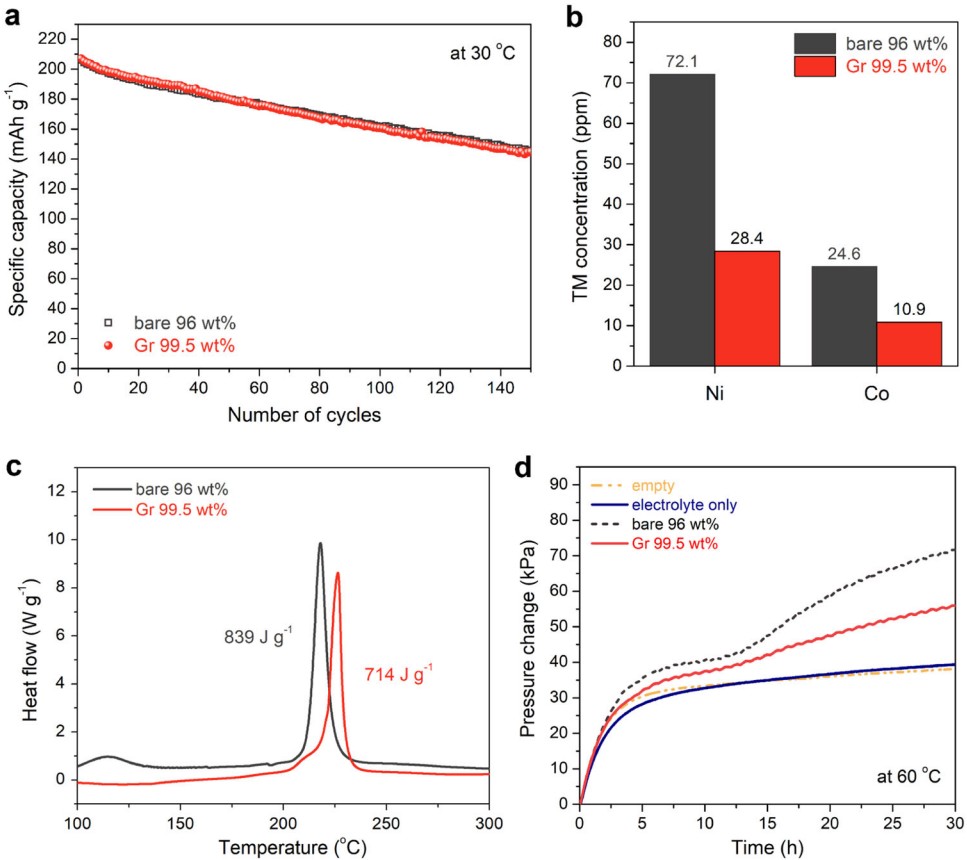

**Fig. 4 Thermal and structural stability of Gr-coated NCA cathodes. a** Discharge capacity of the Gr-coated electrode ($\varphi = 99.5:0:0.5/\rho$ ~3.9 g cm$^{-3}$) and bare electrode ($\varphi = 96:2:2/\rho$ ~3.3 g cm$^{-3}$), measured at 30 °C in the voltage range of 2.75–4.5 V vs. Li$^{+}$/Li at a current rate of 0.5 C. **b–d** Thermal stability tests of Gr-coated and bare electrodes subjected to charging to 4.3 V vs. Li$^{+}$/Li. **b** Content of TM ions dissolved from the electrodes immersed in the electrolyte for one week at 60 °C. **c** DSC profiles. **d** Pressure changes measured using a gas-cell system at 60 °C. For reference, an empty cell and electrolyte-containing cell without any electrode are examined.

electrode shows a high $Q_{\text{areal}}$, 4.9–5.6 mAh cm$^{-2}$ (~37% increment) at low currents, compared to the bare electrode ($Q_{\text{areal}}$ ~3.7–4.0 mAh cm$^{-2}$); however, the $Q_{\text{areal}}$ values cross over at ~2 C (Fig. 5d). The abrupt capacity decreases at high currents are ascribed to the highly dense electrode architecture that limits Li-ion transport through the electrolyte[41]. As shown by the $Q_{\text{vol}}$ plot with respect to the areal current (Fig. 5e), the Gr-coated electrodes deliver considerably higher $Q_{\text{vol}}$ values than those of the bare one at current rates <1 C. For example, at 0.2 C, the Gr-A electrode and Gr-B electrode exhibited $Q_{\text{vol}}$ of ~860 mAh cm$^{-3}$, which is ~34% higher than that of the bare ($Q_{\text{vol}}$ ~640 mAh cm$^{-3}$). The $Q_{\text{vol}}$ of the Gr-A and Gr-B are 790 and 782 mAh cm$^{-3}$, respectively, even at 1 C, while that of the bare is 607 mAh cm$^{-3}$.

We assessed the full-cell cycle performance (NCA|graphite) using the two different NCA cathodes of bare ($\varphi = 96:2:2$, $\rho$ ~3.3 g cm$^{-3}$) and Gr-coated, dense NCA electrodes ($\varphi = 99:0:1$, $\rho$ ~4.3 g cm$^{-3}$) in a pouch-type cell format (Fig. 5f). The $Q_{\text{vol}}$ retention of the two full cells is presented in Fig. 5g, showing similar cycling stability during 200 cycles at 0.5 C (1.7 mA cm$^{-2}$). It is noteworthy that the cell using the Gr-coated cathode show a significantly high $Q_{\text{vol}}$ of ~805 mAh cm$^{-3}$ NCA (~38% increase), compared to that of the cell with the bare cathode ($Q_{\text{vol}}$ of ~583 mAh cm$^{-3}$ NCA) at first cycle. After 200 cycles, the Gr-coated cell still delivered a higher $Q_{\text{vol}}$ of ~666 mAh cm$^{-3}$ NCA (~83% retention) than that of the bare one ($Q_{\text{vol}}$ of ~422 mAh cm$^{-3}$ NCA; ~72% retention; refer to Supplementary Fig. 31).

## Discussion

We have demonstrated an effective method of improving the energy density of Ni-rich oxide cathodes by minimizing redundant portions of large-volume-inactive components, such as CB and PVdF, within the electrodes through conformal Gr coating. Although the Gr-coated, dense electrodes showed poor rate performance at high C-rates (>2 C), when considering a typical battery run times (≥1 h) in practical applications, our electrode design reaches a satisfactory compromise by achieving high $Q_{\text{areal}}$ and high $Q_{\text{vol}}$, as a result of maximizing the amount of active material (~99 wt%). The $Q_{\text{vol}}$ (~860 mAh cm$^{-3}$ at 0.2 C) of the Gr-coated NCA cathode surpasses the $Q_{\text{vol}}$ (760 mAh cm$^{-3}$ at 0.1 C) of Gr-coated LiNi$_{0.6}$Co$_{0.1}$Mn$_{0.3}$O$_2$ (NCM613) cathodes, which were processed through Nobilta milling by Son et al.[17]. They employed electrode parameters of $\varphi$(NCM613:CB:Gr: PVdF) = 97:0.5:1.0:1.5, $m_{\text{areal}}$ ~25 mg cm$^{-2}$, and $\rho$ ~4.1 g cm$^{-3}$. Park et al. demonstrated NCM811 cathodes exhibiting ultrahigh $Q_{\text{areal}}$ with thicknesses of up to ~800 μm using segregated carbon nanotube (CNT) conductive networks[43]. Although direct comparison between the Gr-coated NCA electrodes and their NCM811/CNT electrodes is difficult, our Gr-coated cathodes show meaningful performance metrics, when compared to their data of $Q_{\text{areal}} = 5$–6 mAh cm$^{-2}$ at ~5 mA cm$^{-2}$ (projected from the corresponding graph at $m_{\text{areal}} = 43$–98 mg cm$^{-2}$) and their less dense electrode architectures (≥200 μm in thickness and $\rho \leq$ 3.3 g cm$^{-3}$). Recently, we found a similar study on Gr coatings by Hersam's group, who achieved a Gr-coated NCA cathode ($m_{\text{areal}}$

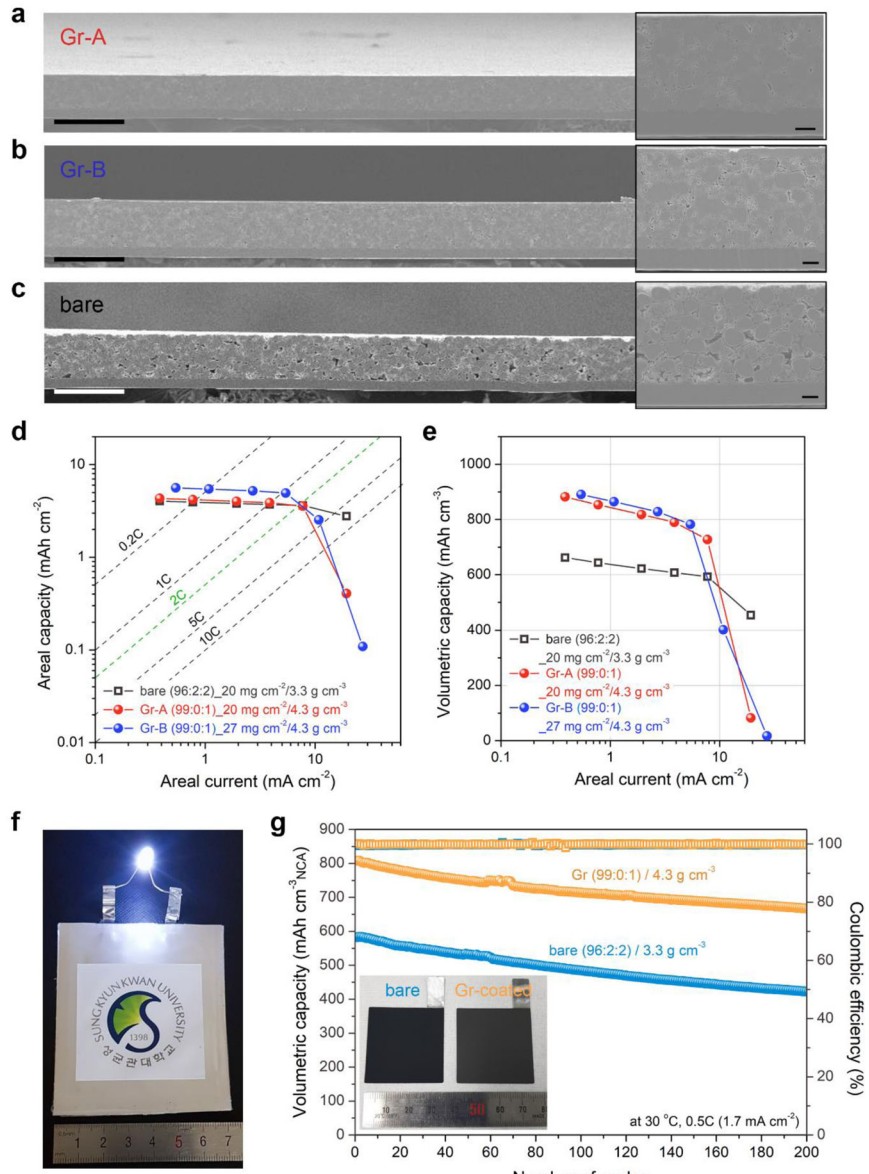

**Fig. 5 Gr-coated, dense NCA cathodes. a–c** Cross-sectional SEM images of the Gr-coated and bare electrodes with different electrode parameters. The Gr-coated electrodes commonly have $\varphi = 99{:}0{:}1$ and $\rho$ ~4.3 g cm$^{-3}$: **a** Gr-A and **b** Gr-B electrodes have ~20 and ~27 mg cm$^{-2}$ in $m_{areal}$, respectively; in contrast, **c** bare electrode has $m_{areal}$ ~20 mg cm$^{-2}$, $\varphi = 96{:}2{:}2$, and $\rho$ ~3.3 g cm$^{-3}$. The scale bars in **a–c** indicate 100 μm and the insets on the right side indicate 10 μm. The electrode thickness is measured to be ~46, ~62, and ~60 μm for Gr-A, Gr-B, and bare electrodes, respectively. Comparison of **d** areal capacity and **e** volumetric capacity of the Gr-coated and bare electrodes with respect to areal current. **f, g** Volumetric capacity of pouch-type full cells using the bare and Gr-coated cathodes ($m_{areal}$ ~17 mg cm$^{-2}$) during repeated cycles at 30 °C and 0.5 C (1.7 mA cm$^{-2}$). The inset in **g** shows a digital photograph of the bare and Gr-coated NCA electrodes, which appear different contrasts in color by different electrode compositions.

~11 mg cm$^{-2}$) with a high $Q_{vol}$ (~675 mAh cm$^{-3}$ at 0.1 C) through a Pickering emulsion coating method without CB[44]. Nonetheless, the performance metrics of our Gr-coated, dense NCA cathodes still exceeded their values and the projected $Q_{vol}$ limit (~690 mAh cm$^{-3}$)[37,44], owing to the conformal Gr coating and highly dense electrode architecture (refer to Supplementary Table 1).

Many studies have used Gr and CNTs to enhance cell performance[18,19,45–48]; however, most of the existing electrode designs do not meet the practical requirements, because of their low tap densities and agglomeration/dispersion challenges[19]. Commendably, our solution-processed Gr coating methodology not only provides sufficient conductive networks within the electrodes and easy processability without notable damage to the active materials, but also widens the viable ranges of electrode density and composition. These improvements pave the way for advanced electrode design toward developing improved LIB cathodes and boost the feasibility of using Gr-based nanotechnology in practical applications.

## Methods

**Electrochemical exfoliation of Gr nanosheets**. We produced Gr nanosheets through the electrochemical exfoliation method (refer to Supplementary Fig. 1)[20,21]. An aqueous solution containing 0.5 M ammonium persulfate $((NH_4)_2S_2O_8)$ was used as the electrolyte, where two graphite foil electrodes were immersed parallel to each other at a fixed distance of 10 mm. When a DC voltage (+10 V) was applied between the graphite electrodes for ~10 min, the positively charged electrode was gradually peeled off to form Gr nanosheets; these were then collected via vacuum filtration followed by washing with DI water and ethanol.

**Preparation of Gr dispersion and coating**. DSPE-mPEG (0–0.2 wt.%, MW 5000, Laysan Bio), an amphiphilic surfactant, was added into 10 ml of the Gr dispersion (1 wt.% Gr in DMF) followed by 30 min of tip sonication, producing small Gr pieces. A total of 3 g of Ni-rich oxides (NCA and NCM811) or LCO powders were added to the Gr solution with a weight ratio of cathode powder:Gr solution = 1: 2. The solution was then subjected to vortex mixing for 5 min. The coated cathode powders were collected by centrifugation at $193 \times g$, followed by washing with ethanol to remove residual Gr and vacuum drying at 150 °C for 5 h. The coating process, including an intermediate 10 min drying step was repeated to enhance Gr coverage. Through controlled experiments, various surfactants/additives, such as Triton-X100 [$(C_{14}H_{22}O(C_2H_4O)_n)$]; nonionic surfactant, Sigma-Aldrich], sodium dodecyl sulfate ($CH_3(CH_2)_{11}SO_4Na$; anionic surfactant, Sigma-Aldrich), cetyl-trimethylammonium bromide ([$(C_{16}H_{33})N(CH_3)_3$]Br; cationic surfactant, Sigma-Aldrich), and PVP (MW 10,000, Alfa-Aesar), were used to study the effect of surfactants on the Gr coating behavior, as compared with DSPE-mPEG.

**Materials characterization**. The surface morphology and microstructure of the cathode powders and electrodes were observed using field-emission SEM (JSM-7600F, JEOL) and Cs-corrected high-resolution TEM (JEM ARM 200 F, JEOL). The electrodes were cross-sectioned using a cross-section polisher (SM-09010, JEOL, USA). The STEM-EELS spectra were acquired using a Gatan GIF Quantum 965 ER under the following conditions: convergence angle ($\alpha$) of 0 mrad, collection angle ($\beta$) of 100 mrad, entrance aperture of 5 mm, energy dispersion of 0.25 eV per channel, energy resolution of 0.1 eV, and acquisition time of 0.1 s. Powder X-ray diffraction (D8 Advance, Bruker, Germany) was measured with Cu K$\alpha$ radiation (=1.5406 Å). To confirm Gr coverage on the NCA particles, confocal Raman mapping was conducted. The Raman shifts were acquired with 80 points per line and 80 lines per image, using a scanning area of $40 \mu m \times 40 \mu m$. The mapping images were constructed with respect to the G band intensity of Gr at 1580 cm$^{-1}$ and the $E_g$ and $A_{1g}$ modes of NCA ~400–600 cm$^{-1}$. The electrical conductivity of powder-type samples was measured using the direct volt–ampere method (MCP-PD51, Mitsubishi Chemical Analytech), in which a sample was pressurized and contacted with a four-point probe. TGA was carried out to evaluate the Gr content in the coated NCA powders at a heating rate of 5 °C min$^{-1}$ from 30 to 1000 °C under atmospheric conditions (TG/DTA7300, Hitachi). The pore volume of the electrodes was measured using an AutoPore V mercury intrusion porosimeter. The surface chemical composition and chemical binding states of Gr were investigated using X-ray photoelectron spectroscopy with an Al K$\alpha$ (1486.68 eV) X-ray source (Thermo Scientific K alpha) and FT-IR spectroscopy (IFS-66/S, TENSOR27, Bruker).

**FM imaging of Gr sheets and Gr-coated NCA particles**. Exfoliated Gr and functionalized Gr dispersions (1 wt% Gr in DMF) were prepared, then diluted with a fluorescence dye solution (RhB, 10$^{-6}$ M in ethanol) using a vortex mixer, and left to stand without any perturbation for 3 h at room temperature. The RhB-treated Gr sheets were acquired by centrifugating and drying, then diluted with an appropriate amount of ethanol again. The labeled Gr solution was dropped onto SiO$_2$/Si wafer and dried at 80 °C for FM imaging. The bare NCA particles (0.2 g) were subjected to vortex mixing in 0.2 g of Gr solutions (1 wt% Gr in DMF) containing 0–0.2 wt% DSPE-mPEG, then were diluted with 20 ml of RhB solution (10$^{-4}$ M in ethanol), and left to stand without any perturbation for 12 h at room temperature. After centrifugation and drying at 80 °C, the RhB-treated NCA powder was spread onto a carbon adhesive tape and planarized with a slide glass for FM imaging. The FM images were obtained using a fluorescence microscope (Olympus corp., BX51M) equipped with green excitation filter (U-MWGS3) and halogen lamp (100 W, U-LH100L-3).

**Electrochemical evaluation**
*Preparation of electrodes and cell assembly (half cells and full cells)*. The NCA and LCO powders and NCM811 powder were supplied by Samsung SDI Co., Ltd. and LANDF Co., Ltd., respectively. The slurries were prepared with 96–99.5 wt.% cathode, 0–2 wt.% CB (Super P), and 0–2 wt.% PVdF binder in 1-methyl-2-pyrrolidone (NMP). The prepared slurry was cast on a 20-µm-thick Al foil using the doctor blade method and dried in a convection oven at 80 °C for 2 h, followed by vacuum drying at 120 °C overnight. The electrode density ($\rho$) was set in the range of 3.3–4.3 ($\pm$0.1) g cm$^{-3}$ by a roll press machine, and the areal mass loading ($m_{areal}$) was ~8 and ~15 mg cm$^{-2}$ for Ni-rich oxides and LCO, respectively. For thick NCA electrodes, the mass loading was increased to 20 ($\pm$1) and 27 ($\pm$1) mg cm$^{-2}$. The electrodes were punched into disks with a diameter of 12 mm. The graphite anode (14 mm in diameter) was prepared using 96 wt% artificial graphite, 1 wt% CB (super P), and 3 wt.% binder comprising styrene-butadiene rubber/carboxymethyl cellulose (1:1) on 20-µm-thick Cu foils. The $\rho$ of graphite anode was set as ~1.5 g cm$^{-3}$. Herein, 2032 coin-type cells were assembled for electrochemical testing in an Ar-filled glove box, with a lithium metal counter-electrode and polypropylene separator (Celgard 2400, Celgard). LiPF$_6$ (1 M) in a mixture solvent of ethylene carbonate, ethyl methyl carbonate, and DMC at a volume ratio of 2:4:4 with 1.5% vinylene carbonate additive was used as the electrolyte.

*Electrochemical testing*. The electrochemical properties were measured using a WBCS3000L battery cycler (WonATech). The formation process was conducted at 0.2 C in the voltage range of 2.75–4.3 V vs. Li$^+$/Li for two cycles with a CC–CV charge using a 0.05 C cutoff. The discharge rate capability was measured at various current rates of 0.1–10 C (1 C = 200 mAh g$^{-1}$ for NCA and NCM811, and 1 C = 150 mAh g$^{-1}$ for LCO) after charging at 0.2 C with a CC–CV protocol using 0.05 C cutoff. The cycle performance was evaluated at 30 and 60 °C in the voltage range of 2.75–4.3 or 2.75–4.5 V vs. Li$^+$/Li, at a current rate of 0.5 C with a CC–CV charging protocol using 0.05 C cutoff after two cycles of formation at 0.2 C. The full cells comprising of NCA cathode and graphite anode at N/P ratio of 1.1 were tested at 30 °C in the voltage range of 2.75–4.2 V at 0.5 C with a CC–CV charging protocol using 0.05 C cutoff condition. The pouch-type full cells were assembled using a NCA cathode (35 mm × 35 mm) paired with an artificial graphite anode (37 mm × 37 mm) at a N/P ratio of 1.1. The stacked electrodes containing a PE separator (43 mm × 43 mm) were vacuum-sealed in an Al pouch while filling the 1 M LiPF$_6$ electrolyte in a dry room with the dew point (less than −70 °C). The pouch cells were aged for 24 h and subjected to formation process in the voltage range of 2.75–4.2 V at 0.1 C (1 C = 3.4 mA cm$^{-2}$ NCA) with a CC–CV charging protocol using 0.05 C cutoff condition, followed by degassing process. After aging further for 24 h, the cells were cycled at 30 °C in the voltage range of 2.75–4.2 V at 0.5 C. The EIS spectra were obtained using a biologic VSP-100 instrument in the frequency range of 1 MHz–1 mHz with a voltage perturbation of 10 mV.

*Evaluation of thermal and structural stability of the charged electrodes*. The sample cathodes were charged to 4.3 V vs. Li$^+$/Li; then, they were disassembled from the half-cell, followed by rinsing with a DMC solvent and drying in an Ar-filled glove box. Thermal behavior of the charged electrodes was analyzed using DSC (Mettler Toledo, DSC 3) from 40 to 300 °C at 5 °C min$^{-1}$ using a high-pressure stainless-steel pan holder with a gold-coated copper seal. The dissolution of TM ions from fully charged cathodes into the electrolyte was quantitatively analyzed by ICP-OES. The pressure change was monitored using a home-made gas-cell system, where the cell contained a charged electrode and fresh electrolyte (10 µl) at 60 °C for 30 h (refer to Supplementary Fig. 28).

## Data availability
The data that support the findings of this study are available from the corresponding author upon reasonable request.

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

## Acknowledgements

This work was supported by a grant from the Korea Evaluation Institute of Industrial Technology (KEIT) funded by the Ministry of Trade, Industry and Energy (MOTIE) [No. 20012341] and the National Research Foundation of Korea (NRF) grant funded by the Ministry of Science and ICT [No. 2020R1C1C1013253].

## Author contributions

C.W.P., S.M.H., and Y.-J.K. conceived the project. C.W.P., J.-H.L., and J.K.S. performed synthesis and coating of graphene. C.W.P., J.K.S., and W.Y.J. performed physical and electrochemical characterizations. J.-H.L. and S.M.H. performed microscopic and spectroscopic characterizations. C.W.P., J.-H.L., and S.M.H. wrote the manuscript. D.W. advised the project, and S.M.H. and Y.-J.K. supervised the project. All authors discussed the results and commented on the manuscript.

## Competing interests

The authors declare no competing interests.
