## [Peer Review File · Nature Communications]

Reviewer #1 (Remarks to the Author):

This work reports that Ni-rich cathodes with Graphene (Gr) coating can enable fabrication of highly packed cathodes with still large amount active materials. A complete set of techniques has been used to characterize the cathodes. The manuscript is interesting and has been written very well, and thus, it is suitable for publication and only requires minor revision. I recommend the authors to address the following issues:

i) The Raman spectrum of Fig. 1(g) on Gr-coated NCA particles shows a carbon black feature around 1320 cm^{-1} . The authors should provide more evidence proving that they have prepared a Gr-coated system and not a CB-coated system.

ii) What is the influence of “Gr-coating” on the well-known crack formation in the microstructure? The charge/discharge-induced microcracks are due to anisotropic lattice parameters change.

Reviewer #2 (Remarks to the Author):

In this manuscript, the authors presented conformal graphene (Gr) coating on Ni-rich oxides enables the fabrication of unprecedentedly highly packed cathodes containing a high content of active material. However, several key issues in the manuscript have problems or not been discussed convincingly as listed below. Therefore, the referee would not recommend this manuscript to be published in Nature Communications.

1. All the electrochemical performance are obtained from 2032 coin-type cells. In practical application, if the cell can be magnified, the as-obtained performance would be instructional. At the same time, the authors in this work always stress on the importance of practical application. Therefore, the authors should present the electrochemical performance with pouch cells or other style (at least 1Ah), not 2032 coin-type cell.

2. From line 112 to 113, there is no big difference between coating with and without DSPE-mPEG in SEM. An extra supporting result should be given to prove the necessities for adding surfactant.

3. From line 82 to 86, a connection between size-reduced Gr and higher-order defects should be given, which will be more logical and convenient for understanding. Besides, as shown in Figure 1a, the Gr coating layer should be rough with separate pieces, clarifying Li ions are transferred through defects not across the basal plane as mentioned above.

4. To investigate the exact location of the surfactant, TEM-EDX on cross-section particle using FIB may be viable.

5. At line 173, a slight drop in Li ions diffusion at high currents for Gr-coated should be mentioned, which further in accordance with the values of Q_{real} for Gr-A and Gr-B decrease at high currents.

6. There are some grammar errors, such as, at line 43, it should be “the state-of-the-art”, in supporting information, at line 172, “t” should be revised as “n”.

Reviewer #3 (Remarks to the Author):

This work coated conformal graphene on $\text{LiNi}_{0.8}\text{Co}_{0.15}\text{Al}_{0.05}\text{O}_2$ and $\text{LiNi}_{0.8}\text{Co}_{0.1}\text{Mn}_{0.1}$ cathode, which was supposed to replace the carbon black as conducting additive. Coating cathode with graphene has been proposed and tested previously by different groups. Although Gr coating of this

work results in a highly dense Ni-rich cathodes (99 wt% NCA) and a high volumetric capacity of ~ 820 mAh cm⁻³, this work does not present a new approach to improve the performance of NMC or NCA but reports a slight increase in the loading and therefore the capacity. I don't think this work has provided the novelty needed to publish in Nature Comm. I think this work should be published in a technical journal such as: Advanced Energy Materials or JPS. A technical suggestion to this paper is that the authors may do some further work for understanding the intrinsic change of Gr coating in terms of altering the diffusion path of lithium ions. For example, the surface reaction can be inhomogeneous with the this type of Gr coating. To find it out, despite the ex situ TEM work in SI, in-situ TEM results can be crucial.

Responses to Referee #1's Comments

This work reports that Ni-rich cathodes with Graphene (Gr) coating can enable fabrication of highly packed cathodes with still large amount active materials. A complete set of techniques has been used to characterize the cathodes. The manuscript is interesting and has been written very well, and thus, it is suitable for publication and only requires minor revision. I recommend the authors to address the following issues:

Reply: We thank the referee for positive comments. According to the comments, we have addressed the issues and improved them appropriately in the revised manuscript.

Comment 1: The Raman spectrum of Fig. 1(g) on Gr-coated NCA particles shows a carbon black feature around 1320 cm^{-1} . The authors should provide more evidence proving that they have prepared a Gr-coated system and not a CB-coated system.

Reply: According to the comment, we have checked the Raman spectrum of Gr-coated NCA particles (Fig. 1g), in comparison to those of the as-exfoliated Gr nanosheets, functionalized Gr nanosheets, and carbon black particles. Typically, in Raman spectra, carbon materials exhibit the G band at $\sim 1580 \text{ cm}^{-1}$ due to an electron-phonon coupling interaction and the D band at $\sim 1350 \text{ cm}^{-1}$ stemming from structural defects. It is well-known that the intensity of the D band relative to that of the G band (I_D/I_G) increases with the amount of disorder. In addition, one extra band with a high intensity at $\sim 2720 \text{ cm}^{-1}$, so-called "2D band", is detected for graphitic carbon materials [1]. The shape and intensity of the 2D band are dependent on the electronic structure and the number of stacked layers in graphitic carbon (refer to *Supporting Data, Fig. S1a'* below). We compared the Raman spectra of the as-exfoliated Gr and functionalized Gr nanosheets with that of carbon black (Super P) particles. As shown in *Supporting Data, Fig. S2'*, the Gr sheets which we fabricated in this work have the relatively strong D and G bands and the 2D band (at $\sim 2700 \text{ cm}^{-1}$), which is obviously distinct from that of the carbon black. When compared to the report by Parvez, K. et al., who fabricated Gr nanosheets using the same electrochemical exfoliation method [2], our XPS and Raman spectra show similar characteristics with their data (refer to *Supporting Data, Fig. S3'*); however, our Gr sheets appear to be of more defective. Thus, these results corroborate that NCA particles were uniformly coated with Gr nanosheets, as described in the manuscript.

With regard to this issue, we have added notation of the Gr bands (D+D' and 2D') in Fig. 1g in the revised manuscript as below.

Fig. S1'. Raman spectra of graphite (a) and disordered carbons (b) recorded at 514 nm. Adapted from Ref. [1].

Fig. S2'. Raman spectra of as-exfoliated Gr and functionalized Gr nanosheets fabricated in this work and carbon black particles.

Fig. S3'. (a) XPS C 1s spectrum and (b) Raman spectrum (excited by 488 nm laser) of a selected exfoliated, bilayer Gr sheet. The ratios of I_D/I_G and I_{2D}/I_G peaks are indicated in the figure. Adapted from Ref. [2]. (c) XPS C 1s spectrum of the as-exfoliated Gr nanosheets fabricated in this work.

Supporting References

[1] Merlen, A., Buijnsters, J. G., & Pardanaud, C. A Guide to and Review of the Use of Multiwavelength Raman Spectroscopy for Characterizing Defective Aromatic Carbon Solids: from Graphene to Amorphous Carbons. *Coatings* **7**, 153 (2017).

[2] Parvez, K. et al. Electrochemically Exfoliated Graphene as Solution-Processable, Highly Conductive Electrodes for Organic Electronics. *ASC Nano* **7**, 3598-3606 (2013).

Change made to the manuscript:

Fig. 1| Gr coating on Ni-rich oxides. a, Schematic of Gr collage on Ni-rich oxide particles. **b-e**, SEM images of Gr-coated (**b,c**) and bare (**e**) NCA particles and HRTEM image of Gr-coated NCA particle (**d**). The scale bar in **b**, **c** and **e**, and **d** indicates 1 μm , 100 nm, and 10 nm, respectively. **f,g**, Optical and confocal Raman mapping images (**f**) and Raman spectrum (**g**) on Gr-coated NCA particles. The scale bars indicate 4 μm . **h**, Electrical conductivity of Gr-coated NCA powder and mixture powders containing NCA and CB with weight ratios of 96:2 and 98:1. **i**,

BET surface area of the bare and Gr-coated NCA powders and the mixture powder (NCA:CB=96:2).

Comment 2: What is the influence of “Gr-coating” on the well-known crack formation in the microstructure? The charge/discharge-induced microcracks are due to anisotropic lattice parameters change.

Reply: As the Reviewer pointed out, Ni-rich oxide cathodes suffer from intraparticle (intergranular) micro-cracking due to the anisotropic structural evolution during lithiation/delithiation processes. As we described in the manuscript, we performed the TEM-EELS analyses of the cycled NCA particles, which were prepared by FIB-cross-sectioning of the NCA electrodes after 100 cycles, in order to investigate the effect of Gr coating on NCA protection during cycling, as shown in *Supporting Data, Fig. S4'* below. Both the bare and Gr-coated NCA particles had a rock-salt-like (NiO) structure in the vicinity of the surface of the primary particles after cycling (refer to the O-K edges of Spot 1,2 in **Fig. S4d'** and **S4i'**). Nevertheless, it appeared that the Gr-coated NCA particles had lesser microcracks than the bare ones. We attributed this difference to the effect of Gr coating, which could alleviate electrolyte penetration into the secondary particle interior to some extent during repeated cycles.

Fig. S4'. TEM images (**a-c, f-h**) and EELS profiles of O-K (**d,i**) and Ni-L_{2,3} (**e,j**) edges for the bare NCA (**a-e**) and Gr-coated NCA (**f-j**) electrodes after 100 cycles at 60 °C. The samples were prepared using a FIB of the corresponding particles. The scale bars in **a,f, b,g,h**; and **c** indicate 2 μm ; 100 nm; and 50 nm, respectively. The EELS profiles of spots 1-3 were acquired from marks 1-3 in **c** and **h**. Both the samples have a NiO (Fm3m)-like surface structure (spot 1-2), while showing the LiNiO₂ with the R3m structure in the inner region (spot 3). It should be noted that the Gr-coated particle (**f**) appears to be relatively intact, as compared to the bare one showing local microcracks (**a**).

Responses to Referee #2's Comments

In this manuscript, the authors presented conformal graphene (Gr) coating on Ni-rich oxides enables the fabrication of unprecedentedly highly packed cathodes containing a high content of active material. However, several key issues in the manuscript have problems or not been discussed convincingly as listed below. Therefore, the referee would not recommend this manuscript to be published in Nature Communications.

Reply: We thank the referee for critical comments. Considering the raised issues, we have improved the manuscript as below.

Comment 1: All the electrochemical performance are obtained from 2032 coin-type cells. In practical application, if the cell can be magnified, the as-obtained performance would be instructional. At the same time, the authors in this work always stress on the importance of practical application. Therefore, the authors should present the electrochemical performance with pouch cells or other style (at least 1Ah), not 2032 coin-type cell.

Reply: In this work, we did not address cell formats for practical applications, but focused on both 'modification of active materials' and 'electrode engineering' to improve the volumetric capacity of Ni-rich layered oxide cathodes, demonstrating the feasibility of Gr-coated Ni-rich cathodes featuring higher volumetric capacities than that of the bare cathode approaching a commercial level of electrode setting. In practice, the cell format (2032 coin-cells) used in this study is a typical platform for electrochemical testing of LIB electrodes, and thus considered to be sufficient for demonstration of feasibility of our approach.

According to the referee comment, we examined cycle performance of full-cells (graphite|NCA) in a pouch-type cell format. We assembled around 40 mAh-full cells containing a Gr-coated, dense (or bare) NCA cathode paired with a graphite anode, where the N/P ratio was set as 1.1. The detailed assembly procedure is described in Methods section. The digital photograph of a pouch-type cell and the resulting volumetric discharge capacity (Q_{vol}) retention of the full-cells are shown in *Supporting Data, Fig. S5a' and S5b'*, respectively. The two-type full-cells using the bare cathode ($\varphi = 96:2:2$, $\rho \sim 3.3 \text{ g cm}^{-3}$) and the Gr-coated cathode ($\varphi = 99:0:1$, $\rho \sim 4.3 \text{ g cm}^{-3}$) show similar capacity retention during repeated cycles at 0.5C ($1.7 \text{ mA cm}^{-2}_{NCA}$). It is noteworthy that the full cell containing the Gr-coated cathode delivered a higher Q_{vol} of $\sim 805 \text{ mAh cm}^{-3}_{NCA}$ ($\sim 38\%$ increase) that that of the counterpart with the bare cathode (Q_{vol} of $\sim 583 \text{ mAh cm}^{-3}_{NCA}$) at 1st cycle. After 200 cycles, the Gr-coated cell had a high Q_{vol} of $\sim 666 \text{ mAh cm}^{-3}_{NCA}$ ($\sim 83\%$ retention), compared to that of the bare one (Q_{vol} of $\sim 422 \text{ mAh cm}^{-3}_{NCA}$; $\sim 72\%$ retention). The voltage profiles of the full-cells for 200 cycles are displayed in **Fig. S5c'** and **S5d'**.

The referee commented demonstration of '1 Ah-level-full cells.' When considering the electrode dimension relative to nominal cell capacities, 1 Ah-level cells necessitate fabrication of uniform electrodes with a dimension of $\sim 200 \text{ cm}^2$ and $\sim 270 \text{ cm}^2$ for Gr-coated and bare NCA cathodes, respectively. Such large-area-electrode formulation and cell assembly require commercial-level of manufacturing facilities with established processing conditions, which are believed to be beyond typical lab-scale formats and our scope of this study.

With regard to this issue, we have added the pouch-type full-cell data and relevant description in Fig. 5 of the revised manuscript and supplementary information (Fig. S31) as below. We thank the referee for critical comments.

Fig. S5'. Cycle performance of pouch-type full-cells using the bare and Gr-coated NCA cathodes. **a**, Digital photograph of a pouch-type full-cell. **b-d**, Volumetric capacity (b) and Voltage profiles (c,d) of full-cells made with the NCA cathode and graphite anode, measured at 30 °C in the voltage range of 2.75-4.2 V at 0.5C (1.7 mA cm^{-2}). The bare and Gr-coated electrodes have different electrode parameters of $\phi = 96:2:2$ vs. $\phi = 99:0:1$ and $\rho \sim 3.3 \text{ g cm}^{-3}$ vs. $\rho \sim 4.3 \text{ g cm}^{-3}$. The m_{areal} of both the cathodes is $\sim 17 \text{ mg cm}^{-2}$. The inset in **b** shows digital photographs of the bare and Gr-coated NCA electrodes; which appear different contrasts in color by different electrode compositions.

Change made to the manuscript:

Fig. 5] Gr-coated, dense NCA cathodes. **a-c**, Cross-sectional SEM images of Gr-coated and bare electrodes with different electrode parameters. The Gr-coated electrodes commonly have $\phi = 99:0:1$ and $\rho \sim 4.3 \text{ g cm}^{-3}$; however, the Gr-A (**a**) and Gr-B (**b**) electrodes have ~ 20 and $\sim 27 \text{ mg cm}^{-2}$ in m_{areal} , respectively; in contrast, the bare electrode has $m_{\text{areal}} \sim 20 \text{ mg cm}^{-2}$, $\phi = 96:2:2$, and $\rho \sim 3.3 \text{ g cm}^{-3}$ (**c**). The scale bars in **a-c** and the insets indicate 100 and 10 μm , respectively. The electrode thickness is measured to be ~ 46 , ~ 62 , and $\sim 60 \text{ }\mu\text{m}$ for Gr-A, Gr-B, and bare

electrodes, respectively. **d,e** Comparison of areal capacity (**d**) and volumetric capacity (**e**) of Gr-coated and bare electrodes with respect to areal current. **f,g** Volumetric capacity of pouch-type full-cells (**f**) using the bare and Gr-coated cathodes ($m_{\text{areal}} \sim 17 \text{ mg cm}^{-2}$) during repeated cycles at 30 °C and 0.5C (1.7 mA cm^{-2}). The inset in **g** shows digital photographs of the bare and Gr-coated NCA electrodes; which appear different contrasts in color by different electrode compositions.

Supplementary Fig. 31| Cycle performance of the full-cells (graphite|NCA) using the bare cathode (a) and Gr-coated NCA cathodes (b). The m_{areal} of the NCA cathodes was $\sim 17 \text{ mg cm}^{-2}$. The full-cells were cycled at 30 °C in the voltage range of 2.75-4.2 V at 0.5C.

We assessed cycle performance of full-cells (NCA|graphite) using the two different NCA cathodes of bare electrode ($\varphi = 96:2:2$, $\rho \sim 3.3 \text{ g cm}^{-3}$) and Gr-coated, dense NCA electrode ($\varphi = 99:0:1$, $\rho \sim 4.3 \text{ g cm}^{-3}$) in a pouch-type cell format (Fig. 5f). The Q_{vol} retention of the two full-cells is presented in Fig. 5g, showing similar cycling stability during 200 cycles at 0.5C (1.7 mA cm^{-2}). It is noteworthy that the cell using the Gr-coated cathode show a significantly high Q_{vol} of $\sim 805 \text{ mAh cm}^{-3}_{\text{NCA}}$ ($\sim 38\%$ increase), compared to that of the cell with the bare cathode (Q_{vol} of $\sim 583 \text{ mAh cm}^{-3}_{\text{NCA}}$) at 1st cycle. After 200 cycles, the Gr-coated cell still delivered a higher Q_{vol} of $\sim 666 \text{ mAh cm}^{-3}_{\text{NCA}}$ ($\sim 83\%$ retention) than that of the bare one (Q_{vol} of $\sim 422 \text{ mAh cm}^{-3}_{\text{NCA}}$; $\sim 72\%$ retention) (refer to Supplementary Fig. 31). **(page 13, line 19 to page 14, line 3)**

“The pouch-type full-cells were assembled using a NCA cathode (35 mm x 35 mm) paired with an artificial graphite anode (37 mm x 37 mm) with a N/P ratio of 1.1. The stacked electrodes containing a PE separator (43 mm x 43 mm) were vacuum-sealed in an Al pouch while filling the 1 M LiPF_6 electrolyte in a dry room with the dew point (less than $-70 \text{ }^\circ\text{C}$). The pouch cells were aged for 24 h and subjected to formation process in the voltage range of 2.75–4.2 V at 0.1C ($1\text{C} = 3.4 \text{ mA cm}^{-2}_{\text{NCA}}$) with a CC-CV charging protocol using 0.05C cut-off condition, followed by degassing process. After aging further for 24 h, the cells were cycled at 30 °C in the voltage range of 2.75-4.2 V at 0.5C. The EIS spectra were obtained using a biologic VSP-100 instrument in the frequency range of 1 MHz to 1 mHz with a voltage perturbation of 10 mV.”

Methods (page 17)

Comment 2: From line 112 to 113, there is no big difference between coating with and without DSPE-mPEG in SEM. An extra supporting result should be given to prove the necessities for adding surfactant.

Reply: As the referee pointed out, it appears to be insufficient for demonstration of the surfactant effect on the coating coverage in the SEM result. As shown in *Supporting Data, Fig. S6'* and *Fig. S7'* below, the Gr-coated NCA particles feature blurry surface morphology upon SEM imaging due to the presence of coated Gr (**Fig. S6'**), which is distinct from those of the bare NCA and coated NCA particles without the surfactant. For the NCA particles subjected to coating process without the use of DSPE-mPEG (**Fig. S7'**), it was observed that some pieces of Gr sheets were attached locally on the surface; however, many of such Gr sheets were attached not in a face-to-face manner.

We also studied the effect of the DSPE-mPEG content on the electrochemical properties of NCA cathodes, as we described in Supplementary Note 3 and Supplementary Fig. 15. The bare NCA particles were subjected to coating process (1 time) using Gr dispersions (1 wt% Gr, DMF) with different DSPE-mPEG contents (0-0.2 wt%), and the electrodes were fabricated without the use of CB (NCA:CB:PVdF (φ) = 99.5:0:0.5) at an electrode density (ρ) of $\sim 3.9 \text{ g cm}^{-3}$. As displayed in *Supporting Data, Fig. S8'*, the bare electrode and coated electrode without the use of DSPE-mPEG show almost similar rate properties at 0.1-5C, although the coated one has a slightly higher capacity at low C-rates. With an increase in the DSPE-mPEG content upon Gr coating, the NCA electrodes show better rate performance: the optimal content is found to 0.1 wt%. These results reveal that the DSPE-mPEG plays a pivotal role in conformal Gr coating on NCA particles.

According to the referee comment, we have replaced the existing SEM images (Supplementary Fig. 7) with new data (**Fig. S7'**), in order to enhance the distinction between particle surfaces coated with and without the surfactant. We thank Reviewer for critical comment.

Fig. S6'. SEM images of Gr-coated (thrice) NCA particles. All the scale bars indicate 100 nm.

Fig. S7'. SEM images of NCA particles subjected to Gr-coating process (thrice) without the use of DSPE-mPEG surfactant. The arrows highlight Gr nanosheets which are attached locally on the NCA surface. Some of the Gr nanosheets are observed to be not attached in a face-to-face manner. All the scale bars indicate 1 μm.

Fig. S8'. Rate capability of bare and Gr-coated (1 time) NCA cathodes with different contents of DSPE-mPEG (0-0.2 wt%). All the electrodes have electrode parameters of $\phi = 99.5:0:0.5$ and $\rho \sim 3.9 \text{ g cm}^{-3}$.

Change made to the manuscript:

Supplementary Fig. 7 | SEM images of NCA particles subjected to Gr-coating process (thrice) without the use of DSPE-mPEG surfactant. The arrows highlight Gr nanosheets which are attached locally on the NCA surface. Some of the Gr nanosheets are observed to be not attached in a face-to-face manner. All the scale bars indicate 1 μm .

Comment 3: From line 82 to 86, a connection between size-reduced Gr and higher-order defects should be given, which will be more logical and convenient for understanding. Besides, as shown in Figure 1a, the Gr coating layer should be rough with separate pieces, clarifying Li ions are transferred through defects not across the basal plane as mentioned above.

Reply: It has been reported that for graphitic carbon, Li-ion diffusion across the basal plane is sluggish ($\sim 10^{-11}$ cm² s⁻¹), relative to that along the basal plane ($\sim 10^{-6}$ cm² s⁻¹) [3], featuring the anisotropic ion transport. Thus, it has been recognized that stacked Gr layers (membrane) allow the ion transport mainly through the defects/holes in the sheet plane and boundaries between Gr sheets [4-6], as schematically illustrated in *Supporting Data, Fig. S9'* below. Based on the anisotropic feature of the ion transport, we considered that our few-layer-Gr coatings could degrade the kinetics for Li-ion mass transfer between the electrolyte and active materials, to some extent, upon charging and discharging of Gr-coated Ni-rich cathodes. In this work, we thus modified the exfoliated Gr sheets using a high-power tip-sonication process, which is expected to result in the size-reduction and defects generation [7,8], which in turn, improve the Li-ion transport characteristics of the Gr coating layer on Ni-rich oxides. According to the referee comment, we have modified relevant description and references in the revised manuscript as below.

As for Fig. 1a, we schematically illustrated the surface morphology of Gr-coated Ni-rich oxide particles with colored-paper-attached, resembling an 'collage art', where different colored papers depict separate pieces of attached Gr nanosheets. We illustrated that the macroscopic surface morphology of Gr-coated particle follows that of Ni-rich oxide particles, since the coated Gr layers have nanometer scales in thickness. In addition, the enlarged image on 'Gr-coated Ni-rich oxides' depicts the bonding configuration between Gr layers and Ni-rich oxide via the surfactant. We thank the Reviewer for valuable comments.

Fig. S9'. Schematic illustration of Li-ion transport via Gr sheets between liquid electrolyte and active material. The Li ions are transported through the sheet boundaries and defects/holes in Gr sheet planes. Small-area-Gr-layers (bottom) allow more facile Li-ion transport than a large-area-Gr-layers architecture (top).

Supporting References

- [3] Persson, K. et al. Lithium Diffusion in Graphitic Carbon. *J. Phys. Chem. Lett.* **1**, 1176-1180 (2010).
- [4] Cheng, C. et al. Ion transport in complex layered graphene-based membranes with tunable interlayer spacing. *Sci. Adv.* **2**, e1501272 (2016).
- [5] Yao, F. et al. Diffusion mechanism of lithium ion through basal plane of layered graphene. *J. Am. Chem. Soc.* **134**, 8646-54 (2012).
- [6] Xu, Y. et al. Holey graphene frameworks for highly efficient capacitive energy storage. *Nat. Commun.* **5**, 4554 (2014).
- [7] Cai, X. et al. Effects of tip sonication parameters on liquid phase exfoliation of graphite into graphene nanoplatelets. *Nanoscale Res. Lett.* **13**, 241 (2018).
- [8] Baig, Z. et al. Investigation of tip sonication effects on structural quality of graphene nanoplatelets (GNPs) for superior solvent dispersion. *Ultrason. Sonochem.* **45**, 133-149 (2018).

Change made to the manuscript:

“It has also been reported that for graphitic carbon, Li-ion diffusion across the basal plane is sluggish ($\sim 10^{-11} \text{ cm}^2 \text{ s}^{-1}$), relative to that along the basal plane ($\sim 10^{-6} \text{ cm}^2 \text{ s}^{-1}$) [25], which implies that Gr coatings hinder Li-ion mass transfer between the electrolyte and active materials upon charging and discharging of Gr-coated Ni-rich cathodes. Typically, stacked Gr layers allow the ion transport mainly through the defects/holes in the sheet plane and the boundaries between Gr sheets [25-27]. The higher-order defects, such as divacancies and holes, with low diffusion barrier heights are preferred for facile Li-ion transport [26-28]. To resolve these issues, we employed an amphiphilic surfactant, 1,2-distearoyl-sn-glycero-3-phosphoethanolamine-N-

[methoxy(polyethyleneglycol) (DSPE-mPEG), as glue for Gr coating, together with a sonication step for Gr size-reduction and defects generation [29,30].” (page 4, line 9-19)

References (page 18)

29. Cai, X. et al. Effects of tip sonication parameters on liquid phase exfoliation of graphite into graphene nanoplatelets. *Nanoscale Res. Lett.* **13**, 241 (2018).
30. Baig, Z. et al. Investigation of tip sonication effects on structural quality of graphene nanoplatelets (GNPs) for superior solvent dispersion. *Ultrason. Sonochem.* **45**, 133-149 (2018).

Comment 4: To investigate the exact location of the surfactant, TEM-EDX on cross-section particle using FIB may be viable.

Reply: As the referee commented, we had tried STEM-EDS analyses on Gr-coated NCA particles, including FIB-cross-sectioned particles. However, because of (probably) the quite low amount and intrinsic instability (upon e-beam irradiation) of the organic surfactant (DSPE-mPEG), we could not find any meaningful trace of the surfactant. Instead, we have conducted fluorescence microscopy (FM) on Gr-coated particles. FM is typically used to observe tiny biomaterials using their selective interactions with fluorescence molecules (dyes). In this work, we selected a cationic dye of rhodamine B (RhB) that fluoresces bright-red, since the DSPE-mPEG surfactant contains the anionic phosphate group. We first checked the attachment of DSPE-mPEG on Gr sheets using as-exfoliated Gr with relatively large lateral sizes. The as-exfoliated Gr sheets samples, which were functionalized with different contents of DSPE-mPEG (0-0.2 wt% in 1 wt% Gr/DMF solution), were prepared for FM imaging. The DSPE-mPEG-treated Gr solutions were diluted with a RhB solution (10^{-6} M) and dropped on SiO₂/Si wafer, followed by drying at 80 °C. The resulting FM images of the as-exfoliated and functionalized Gr samples are shown in *Supporting Data, Fig. S10'*. It should be noted that graphitic carbon is capable of quenching the fluorescence from dye molecules adsorbed on its surfaces via the excited-state energy transfer, significantly decreasing the fluorescence intensity [9-11]. The as-exfoliated Gr sheets appeared dark due to the effective quenching ability (**Fig. S10a'**). After the functionalization process with higher contents of DSPE-mPEG, the Gr sheets fluoresce red from their edge regions (see **Fig. S10c' and S10d'**), which appeared to be a result of the

considerably weakened quenching ability due to the presence of dye-adsorbed DSPE-mPEG in the vicinity of the edges.

Next, we performed FM imaging on the NCA particles that were subjected to 1-time Gr-coating process with and without the use of DSPE-mPEG. As shown in *Supporting Data, Fig. S11'*, the NCA particles processed without the DSPE-mPEG show dark contrast, overall; their surroundings appeared weak red due to remnant RhB salts, which could be incorporated during the washing step. In contrast, the bright-red color is clearly observed on the NCA particles coated using DSPE-mPEG (**Fig. S11b'** and **S11c'**). These results demonstrate that the surfactant is present in Gr-coated particles. Nevertheless, the exact location of the surfactant, whether the inside (between Gr and NCA) or the outside (coated particle surface), is still unclear, due to the limited resolution of the analytical tools we used, which requires in-depth investigation further. Based on our findings, we speculate that the surfactant interacts on the basal plane with the hydrophobic tails in the DSPE while being attached onto the NCA surface via hydrogen bonding between the surface hydroxyl group (–OH) and the hydrophilic mPEG head (see **Fig. S11d'**). In addition, the surfactant could bind to Gr edges by the hydrophilic head via hydrogen bonding with the hydroxyl functional groups of Gr, consequently contributing to a conformal Gr coating on NCA in a face-to-face manner.

With regard to this issue, we have added relevant data and description in the revised manuscript and supplementary information as below. We thank the reviewer for valuable comments.

Supporting References

- [9] Kim, J., Cote, L. J., Kim, F. & Huang, J. Visualizing graphene based sheets by fluorescence quenching microscopy. *J. Am. Chem. Soc.* **132**, 260-267 (2010).
- [10] Sun, Z. et al. Towards hybrid superlattices in graphene. *Nat. Commun.* **2**, 559 (2011).
- [11] Srivastava, S., Senguttuvan, T. D. & Gupta, B. K. Highly efficient fluorescence quenching with chemically exfoliated reduced graphene oxide. *J. Vac. Sci. Technol. B* **36**, 04G104 (2018).

Fig. S10'. FM imaging of as-exfoliated Gr and functionalized Gr nanosheets. Optical (left) and FM (right) images of (a) as-exfoliated Gr and (b-d) functionalized Gr nanosheets with different DSPE-mPEG contents (0.05-0.2 wt%) on SiO₂/Si wafer. Rhodamine B was used as fluorescence dye for FM imaging. The dark regions are due to the absence of the dye or the quenching by Gr sheets. The light blue arrows in b indicate the trace of fluorescence near the Gr edges. All the scale bars indicate 10 μ m.

Fig. S11' | FM images of Gr-coated NCA particles and schematic illustration of the bonding configuration between Gr and Ni-rich oxide. Optical (top) and FM (bottom) images of NCA particles, which were subjected to Gr-coating process (1 time) without the DSPE-mPEG surfactant (a) and using the surfactant of 0.1 wt% (b) and 0.2 wt% (c), placed on a carbon adhesive tape. The Gr layer, DSPE-mPEG, and rhodamine B are illustrated using a ball-and-stick model (drew by Marvin Sketch), where red, grey, ivory, blue, and orange colors indicate oxygen, carbon, hydrogen, nitrogen, and phosphorus, respectively (d). The DSPE-mPEG could bind to the oxide surface by the hydrophilic head via “hydrogen bonding” with the hydroxyl groups of oxide surface, while retaining “π-π interactions” between the basal plane of Gr and the hydrophobic tails. In addition, the surfactant could bind to Gr edges by the hydrophilic head via “hydrogen bonding” with the hydroxyl functional groups of Gr, consequently contributing to a conformal Gr coating in a face-to-face manner. The orange dotted boxes highlight the possible reactions between the cationic rhodamine B and the anionic phosphate group of DSPE-mPEG.

Change made to the manuscript:

Fig. 2 | FM images of Gr-coated NCA particles and schematic illustration of the bonding configuration between Gr and Ni-rich oxide. Optical (top) and FM (bottom) images of NCA particles, which were subjected to Gr-coating process (1 time) without the DSPE-mPEG surfactant (a) and using the surfactant of 0.1 wt% (b) and 0.2 wt% (c), placed on a carbon adhesive tape. The Gr layer, DSPE-mPEG, and rhodamine B are illustrated using a ball-and-stick model (drew by Marvin Sketch), where red, grey, ivory, blue, and orange colors indicate oxygen, carbon, hydrogen, nitrogen, and phosphorus, respectively (d). The DSPE-mPEG could bind to the oxide surface by the hydrophilic head via “hydrogen bonding” with the hydroxyl groups of oxide surface, while retaining “ π - π interactions” between the basal plane of Gr and the hydrophobic tails. In addition, the surfactant could bind to Gr edges by the hydrophilic head via “hydrogen bonding” with the hydroxyl functional groups of Gr, consequently contributing to a

conformal Gr coating in a face-to-face manner. The orange dotted boxes highlight the possible reactions between the cationic rhodamine B and the anionic phosphate group of DSPE-mPEG.

Supplementary Fig. 12| FM imaging of as-exfoliated Gr and functionalized Gr nanosheets. Optical (left) and FM (right) images of (a) as-exfoliated Gr and (b-d) functionalized Gr nanosheets with different DSPE-mPEG contents (0.05-0.2 wt%) on SiO₂/Si wafer. Rhodamine B was used as fluorescence dye for FM imaging. The dark regions are due to the absence of the dye or the quenching by Gr sheets. The light blue arrows in b indicate the trace of fluorescence near the Gr edges. All the scale bars indicate 10 μ m.

Supplementary Fig. 13| Schematic of the bonding configuration of Triton X-100 between Gr nanosheets and Ni-rich oxide particles. The surfactant and Gr layer are illustrated using a ball-and-stick model (drawn by Marvin Sketch), where red, grey, and ivory indicate oxygen, carbon, and hydrogen, respectively. The amphiphilic surfactants bind to the oxide surface by their hydrophilic head (“hydrogen bonding” with the hydroxyl groups of oxide surface), while retaining “ π - π interactions” between the basal plane of Gr and their hydrophobic tails. In addition, the surfactant could bind to Gr edges by the hydrophilic head via “hydrogen bonding” with the hydroxyl functional groups of Gr, consequently contributing to a conformal Gr coating in a face-to-face manner.

“Raman mapping data also suggest that the Gr sheets after sonication of Gr dispersion containing DSPE-mPEG show notably high D/G ratios locally, i.e., higher defect densities, compared with the as-exfoliated Gr sheets (Supplementary Fig. 11). We further investigated the presence of DSPE-mPEG surfactant by fluorescence microscopy (FM), which is typically used to observe biomaterials that selectively interact with fluorescence dyes. Considering that the DSPE-mPEG surfactant contains an anionic phosphate group, we used a cationic dye of rhodamine B (RhB) that fluoresces bright-red (refer to the orange boxes in Fig. 2d). Firstly, we examined the as-exfoliated Gr sheets, which were functionalized with different contents of DSPE-mPEG (0-0.2 wt%; refer to Methods), to check the attachment of DSPE-mPEG on Gr sheets. The DSPE-mPEG/Gr solutions were diluted with a RhB solution and dropped on SiO₂/Si wafer. The resulting FM images of the as-exfoliated and functionalized Gr samples are shown in Supplementary Fig. 12. Graphitic carbon has been reported to quench the fluorescence from dye molecules adsorbed on its surfaces via the excited-state energy transfer, significantly decreasing the fluorescence intensity [31-33]. The as-exfoliated Gr sheets appeared dark due to the effective quenching ability. After the functionalization process with higher contents of DSPE-mPEG, the Gr sheets fluoresce red from their edge regions (see Supplementary Fig. S12c,d),

which is indicative of considerably weakened quenching ability, owing to the presence of dye-adsorbed DSPE-mPEG in the vicinity of the edges.

Next, FM was performed to image the NCA particles that were subjected to Gr-coating process (1 time) with and without the use of DSPE-mPEG. As shown in Fig. 2a, the NCA particles processed without the DSPE-mPEG show dark contrast, overall; their surroundings appeared weak red due to remnant RhB salts, which could be incorporated during the washing step. On the other hand, the bright-red color is clearly observed on the NCA particles coated using DSPE-mPEG (Fig. 2b,c). These results verify that the surfactant is present in Gr-coated particles. We speculate that the surfactant interacts on the basal plane with the hydrophobic tails in the DSPE while being attached onto the NCA surface via hydrogen bonding between the surface hydroxyl group (–OH) and the hydrophilic mPEG head (see Fig. 2d and Supplementary Fig. 13) [23, 28]. In addition, the surfactant could bind to Gr edges by the hydrophilic head via hydrogen bonding with the hydroxyl functional groups of Gr, consequently contributing to a conformal Gr coating on NCA in a face-to-face manner. This conjecture is partly corroborated by controlled experiments on different types of surfactants (refer to Supplementary Note 2 and Supplementary Fig. 14).” **(page 6-7)**

FM imaging of Gr sheets and Gr-coated NCA particles. As-exfoliated Gr and functionalized Gr dispersions (1 wt% Gr in DMF) were prepared and then diluted with a fluorescence dye solution (rhodamine B; RhB, 10^{-6} M in ethanol) using a vortex mixer, followed by keeping them without any perturbation for 3 h at room temperature. The RhB-treated Gr sheets were acquired by centrifugating and drying and then diluted in an appropriate amount of ethanol again. The labeled Gr solution was dropped onto SiO₂/Si wafer and dried at 80 °C for FM imaging. The bare NCA particles (0.2 g) were subjected to vortex mixing in 0.2 g of Gr solutions (1 wt% Gr in DMF) containing 0-0.2 wt% DSPE-mPEG and then diluted with 20 ml of RhB solution (10^{-4} M in ethanol), followed by leaving them without any perturbation for 12 h at room temperature. After centrifugation and drying at 80 °C, the RhB-treated NCA powders were spread onto a carbon adhesive tape and planarized with a slide glass for FM imaging. The FM images were obtained using a fluorescence microscope (Olympus corp., BX51M) equipped with green excitation filter (U-MWGS3) and halogen lamp (100 W, U-LH100L-3). **Methods (page 16)**

References (page 18)

31. Kim, J., Cote, L. J., Kim, F. & Huang, J. Visualizing graphene based sheets by fluorescence quenching microscopy. *J. Am. Chem. Soc.* **132**, 260-267 (2010).
32. Sun, Z. et al. Towards hybrid superlattices in graphene. *Nat. Commun.* **2**, 559 (2011).
33. Srivastava, S., Senguttuvan, T. D. & Gupta, B. K. Highly efficient fluorescence quenching with chemically exfoliated reduced graphene oxide. *J. Vac. Sci. Technol. B* **36**, 04G104 (2018).

Comment 5: At line 173, a slight drop in Li ions diffusion at high currents for Gr-coated should be mentioned, which further in accordance with the values of Q_{areal} for Gr-A and Gr-B decrease at high currents.

Reply: In order to investigate the effect of Gr coating on the Li-ion diffusion kinetics, we conducted cyclic voltammetry (CV) measurements on the bare and Gr-coated NCA electrodes and compared their Li^+ diffusion coefficients (D_{Li^+}), which were estimated based on the Randles-Sevcik equation, as shown in *Supporting Data, Fig. S12'*. Since the peak current (i_p) is proportional to the surface area of redox-active materials (A), Li^+ diffusion coefficient ($D_{\text{Li}^+}^{0.5}$), and scan rate upon CV measurements ($\nu^{0.5}$), the slope ($i_p/\nu^{0.5}$) implies the Li^+ diffusion property. As depicted in **Fig. S12c'**, the Gr-coated electrode showed almost similar slopes to those of the bare electrode, indicating that the Gr coating has no notable negative effect on the Li-ion mass transfer, particularly upon lithiation (discharging). This result is in agreement with the similar discharge rate capabilities of the bare and Gr-coated electrodes, as demonstrated in Fig. 3a of the revised manuscript.

In Fig. 5d (of revised manuscript), we compared the areal capacities (Q_{areal}) of the bare and Gr-coated NCA electrodes with respect to areal current. As the referee indicated, the Gr-A and Gr-B electrodes exhibited inferior Q_{areal} values at high currents, compared to that of the bare one. It should be noted that the high-rate capability of electrodes depends on various factors occurred in electrodes, including Ohmic resistance, charge transfer polarization, ion concentration polarization, and solid-state diffusion polarization. The Gr-A electrode had the same areal mass-loading (m_{areal}), but highly dense electrode architecture ($\rho \sim 4.3 \text{ g cm}^{-3}$), compared to the bare electrode ($m_{\text{areal}} \sim 20 \text{ mg cm}^{-2}$ and $\rho \sim 3.3 \text{ g cm}^{-3}$). The Gr-B electrode featured even more severe electrode parameters ($m_{\text{areal}} \sim 27 \text{ mg cm}^{-2}$ and $\rho \sim 4.3 \text{ g cm}^{-3}$). Carbon black (CB)-free electrode composition ($\varphi = 99:0:1$) aside, such harsh electrode architectures with high ρ and/or high m_{areal} are considered to be mainly responsible for an abrupt decrease in Q_{areal} at high currents for Gr-coated electrodes. We thus concluded that the Gr coating on Ni-rich oxide cathodes has no prominent negative effects on the electrochemical properties, as demonstrated in the revised manuscript (Figs. 3-5). We believe that there is a trade-off between the electrical conductivity and ionic conductivity by Gr coating on Ni-rich cathodes, although the Gr-coating layer, even with extremely small volume/weight, acts as effective conducting agent within the electrodes, enabling fabrication of CB-free, highly dense cathodes with large volumetric capacities.

We thank the referee for important comments.

Fig. S12'. Cyclic voltammograms of (a) bare and (b) Gr-coated NCA electrodes, measured in the voltage range of 2.75-4.3 V vs. Li^+/Li at scan rates of 0.1-1 mV s^{-1} . The peak current (i_p) was determined at peaks related to the phase transition between the hexagonal (H1) and monoclinic (m) structures. The bare electrode has $\phi = 96:2:2$ and $\rho \sim 3.3 \text{ g cm}^{-3}$, whereas the Gr-coated one has $\phi = 99.5:0:0.5$ and $\rho \sim 3.9 \text{ g cm}^{-3}$. (c) Relationship between the peak current (i_p) and the square root of the scan rate ($v^{1/2}$) for lithiation/delithiation.

Comment 6: There are some grammar errors, such as, at line 43, it should be “the state-of-the-art”, in supporting information, at line 172, “t” should be revised as “n”.
Reply: According to the comment, we have corrected the typos in the revised manuscript. We thank Reviewer for helpful comments.

Change made to the manuscript:

“The electrodes in the state-of-the-art LIB cells employ few percentages (wt%) of inactive materials, such as large-surface-area carbonaceous conducting agents and polymeric binders to ensure the conductive percolation network and mechanical endurance, respectively.” (page 2, line 20-23)

“... where σ_0 is a constant related to the conductivity of the filler network and n is the percolation exponent.” (page 12 in Supplementary Note. 1)

Responses to Referee #3's Comments

This work coated conformal graphene on $\text{LiNi}_0.8\text{Co}_0.15\text{Al}_0.05\text{O}_2$ and $\text{LiNi}_0.8\text{Co}_0.1\text{Mn}_0.1$ cathode, which was supposed to replace the carbon black as conducting additive. Coating cathode with graphene has been proposed and tested previously by different groups. Although Gr coating of this work results in a highly dense Ni-rich cathodes (99 wt% NCA) and a high volumetric capacity of $\sim 820 \text{ mAh cm}^{-3}$, this work does not present a new approach to improve the performance of NMC or NCA but reports a slight increase in the loading and therefore the capacity. I don't think this work has provided the novelty needed to publish in Nature Comm. I think this work should be published in a technical journal such as: Advanced Energy Materials or JPS. A technical suggestion to this paper is that the authors may do some further work for understanding the intrinsic change of Gr coating in terms of altering the diffusion path of lithium ions. For example, the surface reaction can be inhomogeneous with the this type of Gr coating. To find it out, despite the ex situ TEM work in SI, in-situ TEM results can be crucial.

Reply: As the referee pointed out, there have been many reports on the use of Gr sheets as conductive agents/coatings in battery electrodes, including Ni-rich oxide cathodes. Nevertheless, the electrode formulations in existing many studies are far from those required for practical implementation in battery industry. This is fundamentally due to the low tap densities and the agglomeration/dispersion issues of Gr. Moreover, most of the reports have examined electrochemical properties of electrodes fabricated under the low electrode densities ($< 3.0 \text{ g cm}^{-3}$) and/or low mass-loadings ($< 8 \text{ mg cm}^{-2}$), leading to a huge gap between academic research and practical applications.

In this work, we show the improvements of electrode performance metrics, such as areal capacity (Q_{areal}) and volumetric capacity (Q_{vol}), by '(active) materials engineering' using a conformal Gr coating, as well as by 'electrode engineering' for practical implementation. We used Gr nanosheets fabricated through electrochemical exfoliation with commercial graphite foils and no post-processing, which is distinct from conventional approaches based on solution-processed reduced graphene oxide, chemical-vapor-deposited-Gr, or mechanical milling processes. Our effective, conformal Gr coating method enables the minimization of the content of inactive materials, such as CB and PVdF binder, and hence the fabrication of unprecedentedly highly packed cathodes containing a high content of active material ($\sim 99 \text{ wt}\%$) and no CB. With the electrode parameters of 99 wt% Gr-coated NCA and electrode density (ρ) of $\sim 4.3 \text{ g cm}^{-3}$, we demonstrate excellent performance metrics: $Q_{\text{areal}} \sim 5.4 \text{ mAh cm}^{-2}$ ($\sim 38\%$ increase) and $Q_{\text{vol}} \sim 863 \text{ mAh cm}^{-3}$ ($\sim 34\%$ increase) at 0.2C ($\sim 1.1 \text{ mA cm}^{-2}$), as compared to the bare electrode approaching a commercial level of electrode setting (96 wt% NCA; $\rho \sim 3.3 \text{ g cm}^{-3}$). These capacity values surpass those of existing reports using Gr coatings or CNTs-network and further projected Q_{vol} limit of NCA [12,13], as discussed in the revised manuscript. For comparison, we summarized literatures of the electrode parameters and measured capacity metrics of Gr-coated cathodes for LIBs in Supplementary Table 1 of the revised Supplementary Information.

In practice, state-of-the-art LIBs employ Ni-rich oxide cathodes fabricated with electrode densities of 3.3~3.4 g cm⁻³, which has been recognized to be close to the upper limit when using nanoscale, large-volume CB conducting agents. We exemplified the microstructural change of NCA electrodes with respect to the electrode density in *Supporting Data, Fig. S13'* below. For the NCA electrode with $\varphi = 96:2:2$ (the left in **Fig. S13'**), with increasing the electrode density from 3.3 g cm⁻³ to 3.8 g cm⁻³, considerable amounts of the NCA particles are easily found to be severely pulverized, due to the high-pressure calendaring process and to the presence of CB particles (refer to the red dotted circles). This indicates that Ni-rich layered oxide cathodes reach the limits in terms of volumetric energy density, as long as they employ CB as conducting additives within the electrodes. On the other hand, for the Gr-coated NCA electrode with $\varphi = 99:0:1$ (the right in **Fig. S13'**), it appears to be relatively intact even with the increasing electrode density to 4.3 g cm⁻³, which is the result of scavenging conventional, large-volume CB in electrodes. In this regard, our findings show scientific novelty, as well as technological advances in the viewpoint of practical applications.

We believe that our Gr coating approach can circumvent the existing agglomeration/dispersion issues and be scalable for mass-production. It will also widen the viable ranges of electrode density and composition in terms of electrode design, bridging the gap between materials engineering and electrode design for improving battery performance. This work addressed not only the synthesis of small Gr nanosheets and coating on oxide cathode particles, but also probed the mechanism behind the Gr coating using surfactants, based on various analytical techniques. Through the revision process in this round, fluorescence microscopy data on Gr-coated particles, which has hardly been found in the fields of batteries and battery materials, was also added, enriching the manuscript and content. Thus, we consider that our work is well-matched with the journal scope and will be of wide interests to the diverse readership.

With regard to the issue of altering the Li-ion diffusion pathways by Gr coating, we addressed the Li-ion diffusion kinetics using CV measurements based on the Randles–Sevcik equation ($i_p \propto A \cdot D_{Li^+}^{0.5} \cdot \nu^{0.5}$), as described in Fig. 3b and Supplementary Fig. 19 of the revised manuscript. It appears that the Gr coating has no prominent effect on the Li-ion mass transfer upon charge (delithiation) and discharge (lithiation). We consider that the ultrathin (small-volume) Gr coatings effectively provide electrically conductive pathways within electrodes, while allowing Li-ion transport through their sheet boundaries and/or higher-order defects (holes). As the referee commented, in-situ TEM studies may be a powerful approach to observe changes of the surface reactions and coating morphology of a Gr-coated NCA particle upon lithiation/delithiation, which, however, is beyond our scope of this work.

With regard to this issue, we have revised relevant description in the revised manuscript as below. We thank the referee for critical comments.

Fig. S13'. Cross-sectional SEM images of the bare and Gr-coated NCA electrodes with different electrode densities. The bare (left) and Gr-coated (right) electrodes have $\phi = 96:2:2$ and $\phi = 99:0:1$, respectively. The red dotted circles highlight pulverized regions within the electrode. All the scale bars indicate 10 μm .

Supporting References

[12] Park, K. *et al.* Concurrently Approaching Volumetric and Specific Capacity Limits of Lithium Battery Cathodes via Conformal Pickering Emulsion Graphene Coatings. *Adv. Energy Mater.* **10**, 2001216 (2020).

[13] Kim, J., Lee, H., Cha, H., Yoon, M., Park, M. & Cho, J. Prospect and Reality of Ni-Rich Cathode for Commercialization. *Adv. Energy Mater.* **8**, 1702028 (2018).

Change made to the manuscript:

“The Q_{vol} ($\sim 860 \text{ mAh cm}^{-3}$ at 0.2C) of the Gr-coated NCA cathode surpasses the Q_{vol} (760 mAh cm^{-3} at 0.1C) of Gr-coated $\text{LiNi}_{0.6}\text{Co}_{0.1}\text{Mn}_{0.3}\text{O}_2$ (NCM613) cathodes, which were processed through Nabilta milling by Son *et al.* [18].” (page 14, line 12-14)

“Quite recently, we found a similar study on Gr coatings by Hersam’s group, who achieved a Gr-coated NCA cathode ($m_{\text{areal}} \sim 11 \text{ mg cm}^{-2}$) with a high Q_{vol} ($\sim 675 \text{ mAh cm}^{-3}$ at 0.1C) through a Pickering emulsion coating method and no use of CB [44]. Nonetheless, the performance metrics of our Gr-coated, dense NCA cathodes still exceed their values and the projected Q_{vol} limit ($\sim 690 \text{ mAh cm}^{-3}$) [9,44], owing to the conformal Gr coating and highly dense electrode architecture (refer to Supplementary Table 1).” (page 14, line 22 to page 15, line 3)

References (page 19)

44. Park, K. *et al.* Concurrently Approaching Volumetric and Specific Capacity Limits of Lithium Battery Cathodes via Conformal Pickering Emulsion Graphene Coatings. *Adv. Energy Mater.* **10**, 2001216 (2020).

Supplementary References

7. Park, K.Y. *et al.* Concurrently Approaching Volumetric and Specific Capacity Limits of Lithium Battery Cathodes via Conformal Pickering Emulsion Graphene Coatings. *Adv Energy Mater.* **10**, 2001216 (2020).
8. He, X. *et al.* Improved Electrochemical Performance of $\text{LiNi}_{0.8}\text{Co}_{0.15}\text{Al}_{0.05}\text{O}_2$ Cathode Material by Coating of Graphene Nanodots. *J. Electrochem. Society* **166**, A1038-A1044 (2019).
9. Son, I.H. *et al.* Graphene balls for lithium rechargeable batteries with fast charging and high volumetric energy densities. *Nat Commun* **8**, 1561 (2017).
10. Shim, J.-H., Kim, Y.-M., Park, M., Kim, J. & Lee, S. Reduced Graphene Oxide-Wrapped Nickel-Rich Cathode Materials for Lithium Ion Batteries. *ACS Appl. Mater. Interfaces* **9**, 18720-18729 (2017).
11. He, J.-r. *et al.* Synthesis and electrochemical properties of graphene-modified $\text{LiCo}_{1/3}\text{Ni}_{1/3}\text{Mn}_{1/3}\text{O}_2$ cathodes for lithium ion batteries. *RSC Adv.* **4**, 2568-2572 (2014).
12. Noh, H.K., Park, H.-S., Jeong, H.Y., Lee, S.U. & Song, H.-K. Doubling the Capacity of Lithium Manganese Oxide Spinel by a Flexible Skinny Graphitic Layer. *Angew. Chem Int Ed.* **53**, 5059-5063 (2014).
13. Fang, X., Ge, M., Rong, J. & Zhou, C. Graphene-oxide-coated $\text{LiNi}_{0.5}\text{Mn}_{1.5}\text{O}_4$ as high voltage cathode for lithium ion batteries with high energy density and long cycle life. *J. Mater. Chem A* **1**, 4083 (2013).
14. Zhou, X., Wang, F., Zhu, Y. & Liu, Z. Graphene modified LiFePO_4 cathode materials for high power lithium ion batteries. *J. Mater. Chem* **21**, 3353-3358 (2011).
15. Wei, W. *et al.* The effect of graphene wrapping on the performance of LiFePO_4 for a lithium ion battery. *Carbon* **57**, 530-533 (2013).

Supplementary Table 1. Literature comparison of the electrode parameters and measured capacities of Gr-coated cathodes for LIBs.

Active materials	Gr type (fabrication method)	Coating method	Electrode composition	Mass loading [mg cm^{-2}]	Electrode density [g cm^{-3}]	Specific capacity [mAh g^{-1}]	Q_{areal} (or Q_{vol})	Ref.
$\text{LiNi}_{0.8}\text{Co}_{0.15}\text{Al}_{0.05}\text{O}_2$	Electrochem. exfoliated Gr	Immersion (amphiphilic surfactants)	AM(Gr):CB:PVDF =98.5(0.5):0:1	~27	~4.3	201@0.2C	$Q_{\text{areal}} \sim 5.4 \text{ mAh cm}^{-2}$; $Q_{\text{vol}} \sim 860 \text{ mAh cm}^{-3}$ @0.2C	Our work
$\text{LiNi}_{0.8}\text{Co}_{0.15}\text{Al}_{0.05}\text{O}_2$	Solution-exfoliated Gr	Pickering emulsion (acetonitrile/hexane)	AM(Gr):CB:PVDF =98.7(0.5):0:0.8	~11	~3.6	~188@0.1C	$Q_{\text{areal}} \sim 2.07 \text{ mAh cm}^{-2}$; $Q_{\text{vol}} \sim 675 \text{ mAh cm}^{-3}$ @0.1C	7
$\text{LiNi}_{0.8}\text{Co}_{0.15}\text{Al}_{0.05}\text{O}_2$	Gr nanodot (electrochem.)	Immersion (ethanol)	AM(Gr):CB:PVDF =79.6(0.4):10:10	1.4	N/A	195@0.1C	$Q_{\text{areal}} \sim 0.27 \text{ mAh cm}^{-2}$ @0.1C	8
$\text{LiNi}_{0.6}\text{Co}_{0.1}\text{Mn}_{0.3}\text{O}_2$	Gr balls (CVD)	Nobilta milling	AM(Gr):CB:PVDF =97.0(1.0):0.5:1.5	~25	~4.1	~196@0.1C	$Q_{\text{areal}} \sim 4.75 \text{ mAh cm}^{-2}$; $Q_{\text{vol}} \sim 760 \text{ mAh cm}^{-3}$ @0.1C	9
$\text{LiNi}_{0.6}\text{Co}_{0.2}\text{Mn}_{0.2}\text{O}_2$	rGO (Hummer's)	Immersion (APTES/toluene)	AM(Gr):CB:PVDF =95.5(0.5):2:2	N/A	N/A	~199@0.1C	N/A	10
$\text{LiCo}_{1/3}\text{Ni}_{1/3}\text{Mn}_{1/3}\text{O}_2$	rGO (Hummer's)	Spray drying (DI water)	AM(Gr):CB:PVDF =72.8(7.2):10:10	3.4-3.6	N/A	224@0.2C	$Q_{\text{areal}} \sim 0.76 \text{ mAh cm}^{-2}$ @0.2C	11
$\text{Li}(\text{Li}_{0.1}\text{Al}_{0.1}\text{Mn}_{1.8})\text{O}_4$	Gr (ball mill of graphite)	High-energy ball-mill	AM:graphite:CB:PVDF =80:7:7:6	N/A	N/A	~100@1C	N/A	12
$\text{LiNi}_{0.5}\text{Mn}_{1.5}\text{O}_4$	rGO (Hummer's)	Immersion (ethanol)	AM(Gr):CB:PVDF =76.2(3.8):10:10	2-3	N/A	~113@0.2C	$Q_{\text{areal}} \sim 0.23 \text{ mAh cm}^{-2}$ @0.2C	13
LiFePO_4	GO (Hummer's)	Spray drying (DI water)	AM(Gr):CB:PVDF =72.8(7.2):15:5	3-4	N/A	148@0.1C	$Q_{\text{areal}} \sim 0.44 \text{ mAh cm}^{-2}$ @0.1C	14
LiFePO_4	rGO (Hummer's)	Immersion (CTAB/DI water)	AM(Gr):CB:PTFE =72.8(7.2):10:10	N/A	N/A	~150@0.1C	N/A	15

Besides, through the revision and review process of manuscript and data in this round, we have found that an error on the thickness measurement of NCA cathodes that were fabricated under high-pressure calendaring process to set high electrode densities. Accordingly, we corrected the film thickness from ~62 μm to ~63 μm , and hence, the estimated electrode density from ~4.4 g cm^{-3} to ~4.3 g cm^{-3} and revised the relevant description (Fig. 5e) in the revised manuscript as below.

“With 99 wt% $\text{LiNi}_{0.8}\text{Co}_{0.15}\text{Al}_{0.05}\text{O}_2$ (NCA) and electrode density of ~4.3 g cm^{-3} , the Gr-coated NCA cathode delivers a high areal capacity of ~5.4 mAh cm^{-2} (~38% increase) and high volumetric capacity of ~863 mAh cm^{-3} (~34% increase) at a current rate of 0.2C (~1.1 mA cm^{-2}).” **(page 2, line 1-3)**

“...we demonstrate highly dense Ni-rich cathodes (99 wt% NCA; electrode density (ρ)~4.3 g cm^{-3}) with a high areal capacity (Q_{areal}) of ~5.4 mAh cm^{-2} (~38% increase) and high volumetric capacity (Q_{vol}) of ~863 mAh cm^{-3} (~34% increase) at current rate of 0.2C (~1.1 mA cm^{-2}).” **(page 3, line 12-15)**

“...the other (denoted as ‘Gr-B’) had an electrode thickness of ~63 μm (m_{areal} ~27 mg cm^{-2}),” **(page 12, line 19-20)**

“The Gr-B electrode shows high Q_{areal} of 4.9-5.6 mAh cm^{-2} (~37% increment) at low currents, compared to the bare electrode (Q_{areal} ~3.7-4.0 mAh cm^{-2}); however, the Q_{areal} values cross over at ~2C (Fig. 5d). The abrupt capacity decreases at high currents are ascribed to the highly dense electrode architecture that limits Li-ion transport through the electrolyte [41]. For the Q_{vol} plot with respect to the areal current (Fig. 5e), the Gr-coated electrodes deliver considerably higher Q_{vol} values than those of the bare one at current rates less than 1C. For example, at 0.2C, the Gr-A electrode and Gr-B electrode exhibit the Q_{vol} of around 860 mAh cm^{-3} , which is ~34% higher than that of the bare (Q_{vol} ~640 mAh cm^{-3}). The Q_{vol} of the Gr-A and Gr-B are 790 mAh cm^{-3} and 782 mAh cm^{-3} , respectively, even at 1C, while that of the bare is 607 mAh cm^{-3} .” **(page 13, line 9-18)**

“The Q_{vol} (~860 mAh cm^{-3} at 0.2C) of the Gr-coated NCA cathode surpasses the Q_{vol} (760 mAh cm^{-3} at 0.1C) of Gr-coated $\text{LiNi}_{0.6}\text{Co}_{0.1}\text{Mn}_{0.3}\text{O}_2$ (NCM613) cathodes, which were processed through Nobilta milling by Son *et al.* [18].” **(page 14, line 12-14)**

We also constructed ‘Discussion’ section while modifying existing ‘Conclusion’ section and added (or revised) a part of description for readers in the revised manuscript as below.

“3 g of Ni-rich oxides (NCA and NCM811) or LiCoO_2 powders.....” **(Methods, page 15)**

In addition, we have corrected some grammatical errors and typos, including “A high Q_{areal} of electrodes...” and “a considerably higher Q_{vol} of ...”.

Reviewer #1 (Remarks to the Author):

The authors have revised the manuscript according to my comments, and I now recommend final acceptance of the manuscript.

Reviewer #2 (Remarks to the Author):

The authors have positively answered my questions. However, it is noted that the principal novelty in this manuscript is using Gr and no conventional conducting agents to realize an enhanced electrochemical performance and a superior areal capacity, which is based on practical implementation. In order to make the substitution plausible and meaningful, a real and effective electrochemical performance of a battery with high capacity is necessary, which is the point that the referee requested a 1 Ah electrochemical performance. Though the authors provided the results from 40 mAh-full cells (the mass loading is ~ 0.2 g which is too low to make the results persuasive), the effectiveness of Gr coating in practical implementation is still not confirmed. Otherwise, the manuscript should provide scientific explanation on how Gr coating works during cycling without conventional conducting agents. For instance, a comprehensive and detailed characterization should be given to illustrate the mechanisms of Li ions diffusion in Gr coated samples, which will make the conclusion more convincing. Therefore, the referee would not recommend this manuscript to be published in Nature Communications.

Reviewer #3 (Remarks to the Author):

I noticed that the authors have carefully revised the manuscript according to the comments of reviewers. For my view, despite that this work still lacks scientific novelty, it indeed reported technical advances in the Gr coating on the cathode side which could benefit the researchers in this field. With this concern in mind, I think the revised manuscript can be published in Nature Communications.

Responses to Referee #1's Comments

Comments: The authors have revised the manuscript according to my comments, and I now recommend final acceptance of the manuscript.

Reply: We thank the referee for positive comments and recommendation.

Responses to Referee #2's Comments

Comments: The authors have positively answered my questions. However, it is noted that the principal novelty in this manuscript is using Gr and no conventional conducting agents to realize an enhanced electrochemical performance and a superior areal capacity, which is based on practical implementation. In order to make the substitution plausible and meaningful, a real and effective electrochemical performance of a battery with high capacity is necessary, which is the point that the referee requested a 1 Ah electrochemical performance. Though the authors provided the results from 40 mAh-full cells (the mass loading is ~0.2 g which is too low to make the results persuasive), the effectiveness of Gr coating in practical implementation is still not confirmed. Otherwise, the manuscript should provide scientific explanation on how Gr coating works during cycling without conventional conducting agents. For instance, a comprehensive and detailed characterization should be given to illustrate the mechanisms of Li ions diffusion in Gr coated samples, which will make the conclusion more convincing. Therefore, the referee would not recommend this manuscript to be published in Nature Communications.

Reply: In this work, we did address both 'modification of active materials (*effective Gr coating*)' and 'electrode engineering (*maximization of electrode density, while minimizing conventional, large-volume conducting agents*)', in order to enhance the volumetric capacity (volumetric energy density) of Ni-rich layered oxide cathodes, rather than 'scale-up (fabrication of large-area electrodes)' of our approach. In practice, there are many reports on improved performance of LIB electrodes in academic research; however, most of them are often inaccessible to practical applications in industry ['Aligning academia and industry for unified battery performance metrics.' *Nat. Commun.* **9**, 5262 (2018)]. This gap between academia and industry stems not from 'lab-scale assessment platforms (small-scale cells, such as coin cells)', but mostly from experimental settings of testing electrodes in labs, such as 'low mass-loading of active materials (<2 mg cm⁻² for cathodes)', as well as 'electrode composition,' i.e., high contents of inactive materials (conducting agents and binders) relative to active materials', which do not make sense for practical use. In this regard, we consider that our work fully meets evaluation standards required for practical implementations and fully demonstrates meaningful scientific and technological advances in relevant research fields.

With regard to the issues raised by the referee, (i) effect of Gr-coating on the electrochemical performance and (ii) Li-ion diffusion mechanism, we have already described relevant findings in the revised manuscript, based on various analytical techniques, such as electron microscopy (Supplementary Fig. 22), cyclic voltammetry (Fig. 3b and Supplementary Fig. 19), DSC and gas-cell pressure measurements (Fig. 4c,d). We found that the conformal Gr coating on Ni-rich oxide cathodes provides sufficient conductive networks within the electrodes without conventional conducting agents, while causing no negative effects on the electrochemical properties and thermal

stability during repeated cycles. Additionally, we compared the CV data of the Gr-coated NCA electrode with $\varphi = 96:2:2$ and $\rho \sim 3.3 \text{ g cm}^{-3}$ (new data) to that of the bare electrode ($\varphi = 96:2:2$ and $\rho \sim 3.3 \text{ g cm}^{-3}$), to explore the effect of Gr coating on the Li-ion diffusion property. The relationship between the peak current (i_p) and the square root of the scan rate ($v^{1/2}$) for lithiation/delithiation of the bare electrode (empty black square) and Gr-coated electrode (filled red sphere) is shown in *Supporting Data, Fig. S1'* below. No difference in the slopes of those plots is found, indicating that the Gr coating has a negligible effect on the Li-ion diffusion behavior within electrodes, which corroborates the positive effect of our Gr coating approach.

With regard to this issue, we have revised Fig. 3b, Supplementary Fig. 19, and relevant description in the revised manuscript as below.

We thank the Reviewer for valuable comments.

Fig. S1'. Relationship between the peak current (i_p) and the square root of the scan rate ($v^{1/2}$) for lithiation/delithiation of bare and Gr-coated NCA electrodes. The bare and Gr-coated electrodes have different electrode parameters of $\varphi = 96:2:2$ vs. $\varphi = 99.5:0:0.5$ and $\rho \sim 3.3 \text{ g cm}^{-3}$ vs. $\rho \sim 3.9 \text{ g cm}^{-3}$.

Change made to the manuscript:

“The comparison between the bare and Gr-coated electrodes with the same electrode parameters ($\phi = 96:2:2$; $\rho \sim 3.3 \text{ g cm}^{-3}$) confirms a negligible effect of the Gr coating on the Li-ion mass transfer.” (page 9, line 5-7)

Fig. 3| Electrochemical performance of Gr-coated NCA cathodes. a, Rate capability of Gr-coated and bare electrodes at 0.1-5C. **b**, Relationship between the peak current (i_p) and the square root of the scan rate ($v^{1/2}$) for lithiation/delithiation. **c**, Discharge capacity of Gr-coated and bare NCA electrodes, measured at 60 °C in the voltage range of 2.75-4.3 V vs. Li⁺/Li at 0.5C. The bare and Gr-coated electrodes have different electrode parameters of $\phi = 96:2:2$ vs. $\phi = 99.5:0:0.5$ and $\rho \sim 3.3 \text{ g cm}^{-3}$ vs. $\rho \sim 3.9 \text{ g cm}^{-3}$. **d**, Areal capacity of full-cells made with the NCA cathode and graphite anode during 300 cycles, measured at 30 °C in the voltage range of 2.75-4.2 V at 0.5C.

Supplementary Fig. 19] Cyclic voltammety curves of bare (**a**) and Gr-coated (**b,c**) NCA electrodes, measured in the voltage range of 2.75-4.3 V vs. Li^+/Li at scan rates of 0.1-1 mV s^{-1} . The peak current (i_p) was determined at peaks related to the phase transition between the hexagonal (H1) and monoclinic (m) structures. The bare electrode (**a**) has $\phi = 96:2:2$ and $\rho \sim 3.3 \text{ g cm}^{-3}$, whereas the Gr-coated one (**b**) has $\phi = 99.5:0:0.5$ and $\rho \sim 3.9 \text{ g cm}^{-3}$. The Gr-coated electrode with $\phi = 96:2:2$ and $\rho \sim 3.3 \text{ g cm}^{-3}$ is also compared (**c**).

Responses to Referee #3's Comments

Comments: I noticed that the authors have carefully revised the manuscript according to the comments of reviewers. For my view, despite that this work still lacks scientific novelty, it indeed reported technical advances in the Gr coating on the cathode side which could benefit the researchers in this field. With this concern in mind, I think the revised manuscript can be published in Nature Communications.

Reply: We thank the referee for positive comments and recommendation.

Besides, we corrected a typo in 'Discussion' section, and revised 'centrifugation speed' in 'Method' section according to the required format. We also removed exaggerated language in the text as below.

"They employed electrode parameters of $\phi(\text{NCM613:CB:Gr:PVdF}) = 97:0.5:1.0:1.5$, $m_{\text{areal}} \sim 25 \text{ mg cm}^{-2}$, and $\rho \sim 4.1 \text{ g cm}^{-3}$." (page 14, line 16-18)

"The coated cathode powders were collected by centrifugation at 193 xg,..." (Method, page 15)

"Here, we show how conformal graphene (Gr) coating on Ni-rich oxides enables the fabrication of ~~unprecedentedly~~ highly packed cathodes containing a high content of active material (~99 wt%) without conventional conducting agents." (page 1, line 21-23)

"Our findings offer a ~~new~~, combinatorial avenue for materials engineering and electrode design toward advanced LIB cathodes." (page 2, line 5-6)

In addition, we have proofread the revised manuscript using a professional service once again. The revised ones are marked in red color.